# Text2PDE: Latent Diffusion Models for Accessible Physics Simulation

**Anthony Zhou, Zijie Li & Amir Barati Farimani** *
Carnegie Mellon University
`{ayz2, zijieli, afariman}@andrew.cmu.edu`

**Michael Schneier & John R. Buchanan, Jr.**
Naval Nuclear Laboratory
`{michael.schneier, jack.buchanan}@unnpp.gov`

## Abstract

Recent advances in deep learning have inspired numerous works on data-driven solutions to partial differential equation (PDE) problems. These neural PDE solvers can often be much faster than their numerical counterparts; however, each presents its unique limitations and generally balances training cost, numerical accuracy, and ease of applicability to different problem setups. To address these limitations, we introduce several methods to apply latent diffusion models to physics simulation. Firstly, we introduce a mesh autoencoder to compress arbitrarily discretized PDE data, allowing for efficient diffusion training across various physics. Furthermore, we investigate full spatio-temporal solution generation to mitigate autoregressive error accumulation. Lastly, we investigate conditioning on initial physical quantities, as well as conditioning solely on a text prompt to introduce text2PDE generation. We show that language can be a compact, interpretable, and accurate modality for generating physics simulations, paving the way for more usable and accessible PDE solvers. Through experiments on both uniform and structured grids, we show that the proposed approach is competitive with current neural PDE solvers in both accuracy and efficiency, with promising scaling behavior up to ~3 billion parameters. By introducing a scalable, accurate, and usable physics simulator, we hope to bring neural PDE solvers closer to practical use.

## 1 Introduction

Neural PDE solvers are an exciting new class of physics solvers that have the potential to improve many aspects of conventional numerical solvers. Initial works have proposed various architectures to accomplish this, such as graph-based (Li & Farimani, 2022; Battaglia et al., 2016), physics-informed (Raissi et al., 2019), or convolutional approaches (Thuerey et al., 2020). Subsequent work on developing neural operators (Li et al., 2021; Lu et al., 2021; Kovachki et al., 2023) established them as powerful physics approximators that can quickly and accurately predict PDEs, and recent work has focused on improving many of their different aspects. These advances have established different models that can adapt to irregular grids (Brandstetter et al., 2023; Li et al., 2023a;c; Wu et al., 2024), speed up model training (Li et al., 2024a; Alkin et al., 2024; Tran et al., 2023; Li et al., 2024b; 2023b), achieve more accurate solutions (Lippe et al., 2023), or generalize to a wide variety of parameters (McCabe et al., 2023; Hao et al., 2024; Herde et al., 2024; Zhou et al., 2024).

Despite these advances, there are still many factors that limit the practical adoption of neural PDE solvers. Great progress has been made in addressing the accuracy and generalizability of neural solvers, however, deploying these advances will require further work in developing new engineering tools. Although neural solvers have become highly capable, they have also become highly specialized; increasingly, the practical use of neural solvers is limited by their inaccessibility and the complexity of the field. Deep learning has already enabled vision and language tools designed with productivity and usability in mind; is there a way we can create similar tools for engineers?

---

*Corresponding Author.

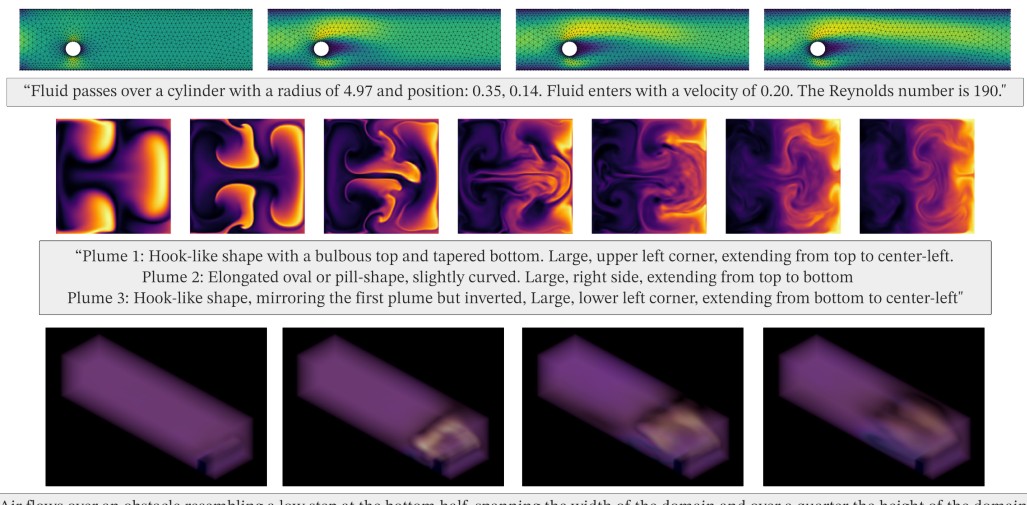

**Figure 1:** We introduce latent diffusion models for physics simulation, with the remarkable ability of generating an entire PDE rollout from a text prompt. Three generated solutions are displayed with their model inputs.

**Text2PDE Models** In this work, we take a step towards creating more accessible physics simulators by introducing language as a novel modality to interact with physics solvers, creating text2PDE models. This has both benefits and drawbacks. One potential benefit is that language is extremely general: while different physical phenomena usually require problem-specific meshing and solvers, humans constantly observe and interact with physics and, as a result, build a common understanding and vocabulary for describing these phenomena. This could reduce the barrier to entry for adopting neural solvers by allowing practitioners to express ideas in language to prompt a physics model. Indeed, the popularity of text2image models is partly due to the ease of expressing visual ideas in language, rather than, for example, needing the computer vision expertise to query a class-conditional image generation model with an ImageNet label. Beyond improving accessibility, language can also have technical benefits. As long as the physical phenomena can be described in text, its domain specialties can be abstracted away, such as the discretization, geometries, or boundaries; this allows models to work with just a single modality. In addition, language can be compact: rather than needing raw physical data to specify a simulation, this information can be distilled into dense textual descriptions.

However, text2PDE models present unique challenges. Since text can describe a broad set of physics, the text2PDE model must generate arbitrarily discretized results. Furthermore, producing high-dimensional spatio-temporal physics simulations from a low-dimensional text prompt can be challenging. Lastly, language can be imprecise in describing many physics problems, and in certain cases, specifying exact initial conditions may be preferred. Indeed, we envision text2PDE models not as a replacement for current methods, but rather as an additional tool for engineers to leverage. We hope to empower engineers with a diverse toolkit, where different neural solvers and even numerical methods can be used together throughout the design cycle to better ideate, simulate, and realize engineering outcomes.

**Contributions** To work towards this goal, we introduce methods to use latent diffusion models to generate PDE solutions. Within this framework, we generate an entire solution trajectory at once to avoid autoregressive error propagation and improve accuracy. We also develop a novel autoencoder strategy to efficiently train/query models and to adapt latent diffusion to arbitrary discretizations. We also investigate conditioning on either text or physics modalities, such as the initial frame of a solution, to switch between an interpretable interface and a more precise one. Lastly, we scale latent diffusion models up to nearly 3 billion parameters to show their ability to accurately generate complex outputs from compact inputs. We believe that this scalability, accuracy, and usability can unlock generative models that can bring neural PDE solvers closer to practical adoption. To further this objective, all code, datasets, and pretrained models are released at: https://github.com/anthonyzhou-1/ldm_pdes

## 2 BACKGROUND

### 2.1 PROBLEM SETUP

We consider modeling solutions to time-dependent PDEs in one time dimension $t = [0, T]$ and multiple spatial dimensions $\mathbf{x} = [x_1, x_2, \ldots, x_D]^T \in \Omega$. Following Brandstetter et al. (2023), we can express these PDEs in the form:

$$\partial_t \mathbf{u} = F(t, \mathbf{x}, \mathbf{u}, \partial_x \mathbf{u}, \partial_{xx} \mathbf{u}, \ldots) \qquad t \in [0, T], \mathbf{x} \in \Omega \tag{1}$$

Where $\mathbf{u} : [0, T] \times \Omega \to \mathbb{R}^{d_p}$ is a quantity that is solved for with physical dimension $d_p$. Furthermore, the PDE can be subject to initial conditions $\mathbf{u}^0(\mathbf{x})$ and boundary conditions $B[\mathbf{u}](t, \mathbf{x}) = 0$:

$$\mathbf{u}(0, \mathbf{x}) = \mathbf{u}^0(\mathbf{x}) \qquad \mathbf{x} \in \Omega \tag{2}$$
$$B[\mathbf{u}](t, \mathbf{x}) = 0 \qquad t \in [0, T], \mathbf{x} \in \partial\Omega \tag{3}$$

In general, the solution $\mathbf{u}$ is discretized at a timestep $t$ to obtain $\mathbf{u}(t, \mathbf{x_m})$ collocated on $M$ mesh points $\mathbf{x}_m \subset \Omega$ for $m \in \{0, 1, \ldots, M\}$. Neural solvers aim to solve time-dependent PDEs by using the solution at a current timestep to predict the solution at a future timestep $\mathbf{u}(t + \Delta t, \mathbf{x_m}) = f_\theta(\mathbf{u}(t, \mathbf{x_m}))$. Many variants of this setup exist, whether it is to use multiple input timesteps or to predict multiple output timesteps. Despite these variants, the dominant approach still follows numerical solvers by autoregressively advancing the solution in time, with certain notable exceptions (Lienen et al., 2024; Du et al., 2024).

Autoregressive solvers are a natural choice for neural solvers; they follow numerical solver precedents, as well as reflect the Markovian nature of time-dependent PDEs. However, this paradigm has many potential drawbacks, namely the stability of autoregressive solvers (Lippe et al., 2023). Although many approaches have been developed to mitigate autoregressive error accumulation (Brandstetter et al., 2023), the fundamental problem still exists: complex PDEs require small timescales to accurately resolve physical phenomena, yet this requires longer rollouts which increase error accumulation (Lozano-Durán & Jiménez, 2014; Lienen et al., 2024). To address this issue, we leverage spatio-temporal diffusion to directly predict an entire solution trajectory. This becomes a generative, rather than autoregressive, problem conditioned on information about the desired PDE solution.

### 2.2 SPATIO-TEMPORAL DIFFUSION FOR PDEs

We consider modeling the conditional probability distribution $p$ of the discretized solution $\mathbf{u} \in \mathbb{R}^{T \times M \times d_p}$ at all timesteps $t \in [0, T]$ and spatial coordinates $\mathbf{x}_m \subset \Omega$ given the initial condition $\mathbf{u}^0$ and boundary condition $B[\mathbf{u}]$:

$$p_\theta(\mathbf{u}|\mathbf{u}^0, B[\mathbf{u}]) \approx p(\mathbf{u}|\mathbf{u}^0, B[\mathbf{u}]) \tag{4}$$

The distribution is approximated using a denoising diffusion probabilistic model (DDPM) $p_\theta$ (Ho et al., 2020; Sohl-Dickstein et al., 2015). DDPMs act to reverse a noising process $q(\mathbf{x}_n|\mathbf{x}_{n-1})$ that transforms a PDE roll-out $\mathbf{u} = \mathbf{x}_0$ over $N$ steps into Gaussian noise (i.e. $q(\mathbf{x}_N|\mathbf{x}_0) \approx \mathcal{N}(0, I)$). New PDE solutions can then be generated by sampling a noise vector $\mathbf{x}_N \sim \mathcal{N}(0, I)$ and iteratively approximating the denoising distribution $p_\theta(\mathbf{x}_{n-1}|\mathbf{x}_n)$. While this generates a random sample from the distribution of all PDE samples ($\mathbf{x}_0 \sim p(\mathbf{u})$), the trajectory of a single solution $\mathbf{u}$ is highly dependent on its initial condition $\mathbf{u}^0$ and boundary condition $B[\mathbf{u}]$. Therefore, the conditional denoising process $p_\theta(\mathbf{x}_{n-1}|\mathbf{x}_n, \mathbf{u}^0, B[\mathbf{u}])$ is used to model the conditional distribution in Equation 4.

**Latent Diffusion** Since $\mathbf{u}$ can be extremely high-dimensional, we perform the forward and reverse diffusion process in a learned latent space. More specifically, given a solution $\mathbf{u} \in \mathbb{R}^{T \times M \times d_p}$, where $T$ represents the temporal resolution, $M$ represents the spatial resolution, and $d_p$ is the physical dimension, an encoder $\mathcal{E}$ can embed $\mathbf{u}$ into a latent vector $\mathbf{z} = \mathcal{E}(\mathbf{u})$. Importantly, $\mathbf{z}$ is in a compressed latent space $\mathbb{R}^{L \times d_l}$, where $L$ is a latent spatio-temporal resolution smaller than $T \times M$, and $d_l$ is the latent dimension. To project latent vectors back to the physical space, a decoder $\mathcal{D}$ is used to reconstruct a sample $\mathbf{u} = \mathcal{D}(\mathbf{z})$.

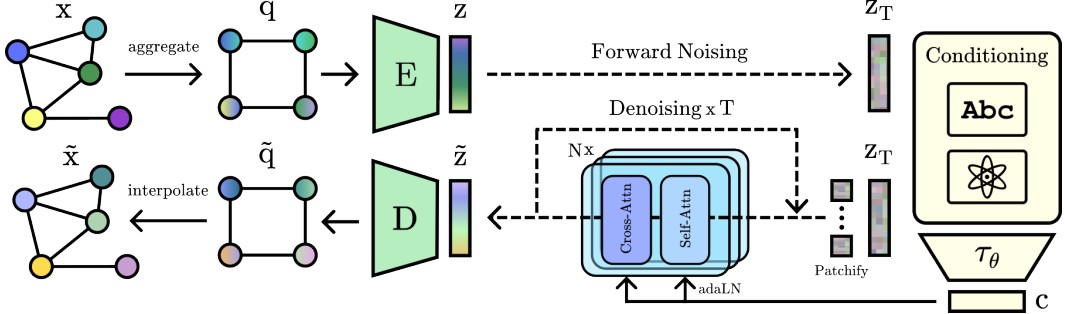

**Figure 2:** The proposed architecture. Samples are mapped to a grid through a learned aggregation before being encoded to a latent vector and noised. A denoising process is learned, with conditioning from text or physics-based modalities. Denoised latents are decoded to a grid and mapped to a mesh through a learned interpolation.

## 3 METHODS

In this section, we describe the main components of using latent diffusion models for physical simulation. An overview of the methods can also be seen in Figure 2.

### 3.1 AUTOENCODERS FOR PDE DATA

Practical physics applications often discretize solutions on an unstructured mesh, which helps improve the accuracy at regions of interest. This can be difficult for conventional autoencoders to accommodate, and indeed much of the previous work is focused on compressing uniformly structured images, videos, or PDE solutions (He et al., 2021; Feichtenhofer et al., 2022; Zhou & Farimani, 2024). Therefore, we propose an autoencoder that can compress data given at arbitrary collocation points and reconstruct them at arbitrary query points to extend diffusion models to unstructured data.

**Mesh Encoder** Following the problem setup, at every timestep $t \in [0, T]$ we can consider $M$ coordinates $\mathbf{x}_m \subset \Omega$ for $m \in \{0, 1, \ldots, M\}$, where a solution vector $\mathbf{u}(t, \mathbf{x}_m) \in \mathbb{R}^{d_p}$ is also defined. This framework captures solutions defined at any point in space and time. We map input timesteps $t$, coordinates $\mathbf{x}_m$, and solutions $\mathbf{u}(t, \mathbf{x}_m)$ onto $T_l$ uniform timesteps $t_l \in [0, T_l]$, $M_l$ uniform grid coordinates $\mathbf{x}_l \subset \Omega$ for $l \in \{0, 1, \ldots, M_l\}$, and latent vectors $\mathbf{q}(t_l, \mathbf{x}_l) \in \mathbb{R}^{d_p}$ at each uniform spatio-temporal coordinate. This is done with the following kernel integral:

$$\mathbf{q}(t_l, \mathbf{x}_l) = \int_{[0,T] \times \Omega} \kappa(t_l, \mathbf{x}_l, t, \mathbf{x}) \mathbf{u}(t, \mathbf{x}) dt d\mathbf{x} \tag{5}$$

In practice, following Li et al. (2020) and Li et al. (2023c), the domain of integration is truncated to a spatio-temporal ball with radius $r$ centered at $t_l$ and $\mathbf{x}_l$, defined as $B_r(t_l, \mathbf{x}_l)$, which enforces a constraint that latent vectors can only depend on local solutions and coordinates. Furthermore, the kernel $\kappa$ is parameterized with a network that computes a kernel value given a physical coordinate $\mathbf{y} = concat(t, \mathbf{x})$ and a latent coordinate $\mathbf{y}_l = concat(t_l, \mathbf{x}_l)$. Lastly, the integral is approximated by a Riemann sum over the $M_b < M$ physical coordinates that lie within the ball $B_r(\mathbf{y}_l)$, denoted as $\mathbf{y}_b \subset [0, T] \times \Omega$ for $b \in \{0, 1, \ldots, M_b\}$. These approximations result in the following expression:

$$\mathbf{q}(\mathbf{y}_l) = \int_{B_r(\mathbf{y}_l)} \kappa(\mathbf{y}_l, \mathbf{y}) \mathbf{u}(\mathbf{y}) d\mathbf{y} \approx \sum_b^{M_b} \kappa(\mathbf{y}_l, \mathbf{y}_b) \mathbf{u}(\mathbf{y}_b) \mu(\mathbf{y}_b) \tag{6}$$

where $\mu(\mathbf{y}_b)$ is the Riemann sum weight for every point $\mathbf{y}_b$ in the ball $B_r(\mathbf{y}_l)$. This kernel integral essentially aggregates neighboring physical solutions for every latent coordinate and has the nice property of approximating an operator mapping from the input function $\mathbf{u}(\mathbf{y})$ to the latent function $\mathbf{q}(\mathbf{y}_l)$. Additional details and visualizations of this kernel integral are provided in Appendix B.3.

**Convolutional Backbone**  Once a uniform latent grid $\mathbf{q}(\mathbf{y}_l)$ is calculated, we can leverage conventional convolutional neural network (CNN) autoencoders to compress and reconstruct the data, which consists of various convolutional, upsampling, downsampling, and attention layers. Specifically, given a latent grid $\mathbf{q} \in \mathbb{R}^{T_l \times M_l \times d_p}$, the CNN encoder will downsample the resolution by a factor of $f$ to produce a latent vector $\mathbf{z} \in \mathbb{R}^{T_l/f \times M_l/f \times d_l}$, where $d_l$ is the latent dimension. The CNN decoder reverses this process to decode a latent grid $\mathbf{q_d}$ from the latent vector $\mathbf{z}$.

Using this method allows our autoencoder to benefit from architecture unification, where best practices from separate domains can benefit PDE applications. In particular, we can leverage variational or vector quantized latent spaces (Kingma & Welling, 2022; Esser et al., 2021) to avoid arbitrarily high-variance latent spaces during training. Furthermore, to improve reconstruction quality, the autoencoder can benefit from incorporating a generative adversarial network (GAN) or perceptual loss metric (Rombach et al., 2022; Zhang et al., 2018). We leave further details and ablation studies of these modifications to Appendix B.1. As a summary, we find that GANs and perceptual guidance can in certain cases improve reconstruction performance; however for simplicity and to ensure the broad applicability of our framework, we omit them in our main results since they require extensive optimization and are highly problem-dependent. Furthermore, we find that highly regularized latent spaces reduce downstream diffusion performance due to lower reconstruction accuracy, and that diffusion backbones are sufficiently parameterized to model higher-variance latent spaces. As a result, our main results use a small Kullback–Leibler (KL) penalty during autoencoder training.

**Mesh Decoder**  To move from the latent grid to the physical mesh, the kernel integral can be reversed. Specifically, given a decoded latent grid $\mathbf{q}_d(\mathbf{y}_l)$, the decoded solution can be calculated by the Riemann sum: $\mathbf{u}_d(\mathbf{y}) = \sum_b^{M_b} \kappa(\mathbf{y}, \mathbf{y}_b) \mathbf{q}_d(\mathbf{y}_b) \mu(\mathbf{y}_b)$. Note that in this case, the sum is evaluated over the ball $B_r(\mathbf{y})$, and the coordinates $\mathbf{y}_b$ are defined as the latent grid points $\mathbf{y}_l$ that lie within $B_r(\mathbf{y})$. This reverse process essentially aggregates neighboring latent vectors for every physical coordinate, and also has the property of being discretization-invariant. In particular, the latent solution can be decoded onto an arbitrary mesh by evaluating the Riemann sum at arbitrary query points $\mathbf{y}$, shown in Figure 8. Lastly, for PDE problems that are uniformly discretized, the kernel integration can be omitted and a CNN autoencoder can be directly used.

**Comparison to GNNs and Neural Fields**  To demonstrate the utility of our method, we draw a comparison with previous work on processing irregular PDE data, namely using GNN or neural field approaches (Pfaff et al., 2021; Cao et al., 2023; Immordino et al., 2024; Yin et al., 2023; Serrano et al., 2024). Although these works do not explicitly train autoencoders, we extend these works to construct GNN- and neural field-based autoencoders to benchmark our proposed method. We provide additional details on these alternative methods as well as present auxiliary results in Appendix B.2. As a summary, we believe that the lack of inductive bias in GNNs and neural fields makes it challenging to aggressively compress and reconstruct unstructured data, which already have very little bias. The proposed approach benefits from the locality constraints of the kernel integral as well as the translation invariance, compositionality, and local connectivity of CNN kernels.

## 3.2  LATENT DIFFUSION

Once a latent space has been learned, we consider freezing the encoder and decoder to learn the latent distribution $p(\mathbf{z})$. Specifically, we train a denoising model $\epsilon_\theta(\mathbf{z}_n, n)$, where $n \in \{0, 1, \dots, N\}$ is an intermediate step within the Gaussian noising process of length $N$, and $\mathbf{z}_n$ is the noised latent at step $n$ [1]. This model is trained by minimizing a reweighted variational lower bound on $p(\mathbf{z})$: $L = \mathbb{E}_{\mathbf{z}, \epsilon \sim \mathcal{N}(0, \mathbf{I}), n} \| \epsilon - \epsilon_\theta(\mathbf{z}_n, n) \|_2^2$. Since the latent $\mathbf{z}_0$ is encoded from a uniform latent grid, we can leverage a Unet to parameterize $\epsilon_\theta$ (Ronneberger et al., 2015). However, recent work by Peebles & Xie (2023) suggests that this inductive bias is not necessary and we evaluate the use of a diffusion transformer (DiT). We find that the DiT backbone outperforms the Unet backbone; the main results are reported with a DiT implementation, but a comparison can be found in Appendix C.2.

Additionally, recent advances in diffusion frameworks have proposed many modifications to the denoising process, the most straightforward of which is scaling the latent space (Rombach et al., 2022).

---

[1]We deviate from the usual notation of $t$ representing the noising timestep to avoid overloading it with $t$ which is previously defined as physical time of the PDE solution $\mathbf{u}(t, \mathbf{x})$.

Furthermore, Nichol & Dhariwal (2021) has shown the potential benefits of using a cosine noise scheduler or learning the variance $\Sigma_\theta(\mathbf{z}_n, n)$ of the reverse process. Lastly, Salimans & Ho (2022) introduced reparameterizing the reverse process to predict the velocity $v = \sqrt{\bar{\alpha}_n}\epsilon - \sqrt{1 - \bar{\alpha}_n}\mathbf{x}_0$, with $\bar{\alpha}_n$ defined as in Ho et al. (2020). We implement and evaluate the effects of these modifications; for interested readers, the findings can be found in Appendix C.1. In general, we find that these modifications can have varying effects on the performance of our model; however, we leave optimizing the diffusion process to future work. For our main results, we scale the latent space, use a linear noise schedule, do not learn the variance, and use an $\epsilon$ parameterization.

## 3.3 Conditioning Mechanisms

**First Frame Conditioning**    To sample from the conditional distribution $p(\mathbf{u}|\mathbf{u}^0, B[\mathbf{u}])$, we add the conditional information to the denoising model $\epsilon_\theta(\mathbf{z_n}, n, \mathbf{u}^0, B[\mathbf{u}])$. For current PDE benchmarks and in many practical applications, the boundary conditions do not change between solutions; as such, we use $\mathbf{u}(0, \mathbf{x}_m)$ as a proxy for both $\mathbf{u}^0$ and $B[\mathbf{u}]$. As such, conditioning information can be added to the denoising backbone by defining a domain-specific encoder $\tau_\theta$ that maps the first frame of the solution to a conditioning sequence $\mathbf{c} = \tau_\theta(\mathbf{u}(0, \mathbf{x}_m)) \in \mathbb{R}^{N_c \times d_c}$, where $N_c$ is the sequence length, and $d_c$ is the conditioning dimension. Conveniently, the proposed encoder $\mathcal{E}$ is agnostic to the temporal and spatial discretization of PDEs; therefore, the same architecture can be used for $\tau_\theta$ by flattening the downsampled latent dimension $(T_l/f \times M_l/f = N_c)$.

**Text Conditioning**    Another interesting conditioning domain is natural language. Although imprecise, humans describe and understand physics through language, and in principle, the initial and boundary conditions of a physics simulation could also be described by text. An advantage of this modality is that it is interpretable and compact; instead of needing to specify the discretization scheme and the physical values at every mesh point, only the important behavior driving the physical phenomena need to be described. However, a clear drawback is the underdetermined nature of the PDE problem: for a given prompt describing the initial and boundary conditions, there may exist many plausible solutions. Furthermore, this approach introduces the problem of captioning physics simulations; we develop additional methods to achieve this and interested readers are directed to Appendix D for details on PDE captioning.

Regardless, text-conditioned PDE simulators are a promising direction and to achieve this, we propose a text-specific encoder $\tau_\theta$ based on pretrained transformer encoders (Vaswani et al., 2023; Devlin et al., 2019). Specifically, given a tokenized and embedded prompt $\mathbf{p} \in \mathbb{R}^{N_c \times d_c}$, with sequence length $N_c$ and dimension $d_c$, a transformer encoder can produce a conditioning sequence $\tau_\theta(\mathbf{p}) = \mathbf{c} \in \mathbb{R}^{N_c \times d_c}$. This has the additional benefit of leveraging large, pretrained transformer encoders; in particular, we fine-tune RoBERTa (Liu et al., 2019) to accelerate the learning of $\tau_\theta$.

**Implementation**    After defining two conditioning modalities and their corresponding encoders $\tau_\theta$, we consider mechanisms for incorporating the conditioning sequence $\mathbf{c}$ into the denoising backbones. For the Unet backbone, a cross-attention layer can be used between a flattened, hidden Unet representation $\varphi(\mathbf{z}_n)$ and the conditioning sequence $\mathbf{c}$ (Rombach et al., 2022). For the DiT backbone, we apply mean pooling to produce $\bar{\mathbf{c}} = mean(\mathbf{c}) \in \mathbb{R}^{d_c}$ in order to apply the *adaLN-Zero* conditioning proposed by Peebles & Xie (2023). Since mean pooling provides only global information, we also insert a cross-attention layer after self-attention DiT layers to allow hidden DiT representations $\varphi(\mathbf{z}_n)$ to attend to the conditioning sequence $\mathbf{c}$ (Chen et al., 2024).

A final consideration is the use of classifier-free guidance to yield a tradeoff between sample diversity and quality (Ho & Salimans, 2022). For PDEs, there is only one solution for a given initial and boundary condition; as a result, the desired behavior is to enhance the sample quality at the cost of diversity. We study this effect for the conditional generation of PDE samples and report findings in Appendix D.4. We find that different weights $w$ only noticeably affect generated samples at their extrema ($w \approx 0, w > 10$); as a result, classifier-free guidance is omitted from our main results.

## 4 Experiments

We explore the proposed latent diffusion model (LDM) for three PDE datasets: 2D flows around a cylinder (Pfaff et al., 2021), 2D buoyancy-driven smoke flows (Gupta & Brandstetter, 2022), and 3D

| Model | Params | Tflops | L2 Loss |
|-------|--------|--------|---------|
| GINO | 72M | 0.73 | 0.2445 |
| MGN | 101M | 32.16 | 0.2617 |
| OFormer | 131M | 17.34 | 0.3386 |
| LDM$_S$-FF | 198M | 0.81 | 0.1522 |
| LDM$_M$-FF | 667M | 1.16 | **0.1309** |
| LDM$_S$-Text | 313M | 0.83 | 0.1796 |
| LDM$_M$-Text | 804M | 1.20 | 0.1476 |

**Table 1: Cylinder Flow.** *Left:* Model parameters, flops, and validation losses across different baselines. *Right:* Sample rollouts for first frame and text conditioned models compared to the reference solution.

turbulence around geometric objects (Lienen et al., 2024). The cylinder flow dataset is discretized on a mesh to smoothly model the cylindrical geometry and its wake; however, the smoke buoyancy and 3D turbulence datasets are uniformly discretized. Additional dataset details can be found in Appendix E. For each model, we report its parameter count and relative L2 loss. Since parameter count can be a poor measure of training cost and model complexity (Peebles & Xie, 2023), we additionally report the floating point operations per second (flops) needed for a single forward pass during training with a batch size of 1, calculated using DeepSpeed (Rasley et al., 2020). Additional details on training and model hyperparameters can be found in Appendix F.

## 4.1 CYLINDER FLOW

**Dataset** Following Pfaff et al. (2021), we use 1000 samples for training and 100 samples for validation. Each sample contains approximately 2000 mesh points and is downsampled to evolve over 25 timesteps, which is compressed to a latent size of $16 \times 16 \times 16$. Each sample is varied in the position and radius of the cylinder, as well as in the inlet velocity of the fluid. This results in samples with Reynolds numbers of 100-1500 and includes Karman vortex streets (Stringer et al., 2014b). In addition, the physical variables are velocity and pressure. Lastly, to caption this dataset, we extract the cylinder position, radius, and Reynolds number and procedurally generate prompts based on a template; we include further details and ablation studies in Appendix D.1.

**Results** We consider benchmarking against GINO (Li et al., 2023c), MeshGraphNet (MGN) (Pfaff et al., 2021), and OFormer (Li et al., 2023a), which represent a variety of operator-, graph-, and attention-based neural solvers. Since these are autoregressive models, we provide the initial frame of the solution and autoregressively predict the next 24 frames. We compare this to our latent diffusion model conditioned on both the first frame (-FF) and a text prompt (-Text), which are trained to generate a full solution trajectory at once. Furthermore, we consider two model sizes: Small (S) and Medium (M); the results are shown in Table 1 with relative L2 loss.

We observe that the latent diffusion model can outperform current models on this benchmark and pressure errors are around the same as velocity errors. Notably the inclusion of flops allows for additional insight into model efficiency. We reproduce results showing that GINO models are very efficient (Li et al., 2023c), yet LDMs can still match this by using a latent space and diffusing an entire rollout at once. Furthermore, our model can achieve lower errors than baselines and more efficiently than graph- or attention-based neural surrogates. Lastly, conditioning on the first frame outperforms text conditioning, presumably because more information is provided in the initial velocity and pressure fields. Text remains an accurate conditioning modality; moreover, the latent diffusion model shows good scaling behavior, with increased performance at larger model sizes.

## 4.2 BUOYANCY-DRIVEN FLOW

**Dataset** We additionally evaluate our model on a regular-grid benchmark of conditional buoyancy-driven smoke rising in a square domain (Gupta & Brandstetter, 2022). We use 2496 training samples and 608 validation samples, with varying initial conditions and buoyancy factors, and at a spatial resolution of $128 \times 128$ and 48 timesteps, which is compressed to a latent dimension of $6 \times 16 \times$

| Model | Params | Tflops | L2 Loss |
|---|---|---|---|
| FNO | 510M | 0.85 | 0.5126 |
| Unet | 580M | 21.48 | 0.3050 |
| Dil-Resnet | 33.2M | 58.75 | 0.4466 |
| ACDM | 404M | 51.23 | 0.4766 |
| $LDM_S$-FF | 243M | 1.41 | 0.3459 |
| $LDM_M$-FF | 725M | 1.68 | 0.3177 |
| $LDM_L$-FF | 1.55B | 2.19 | **0.2728** |
| $LDM_S$-Text | 334M | 1.39 | 0.4290* |
| $LDM_M$-Text | 825M | 1.65 | 0.3158* |
| $LDM_L$-Text | 2.69B | 2.53 | 0.2944* |

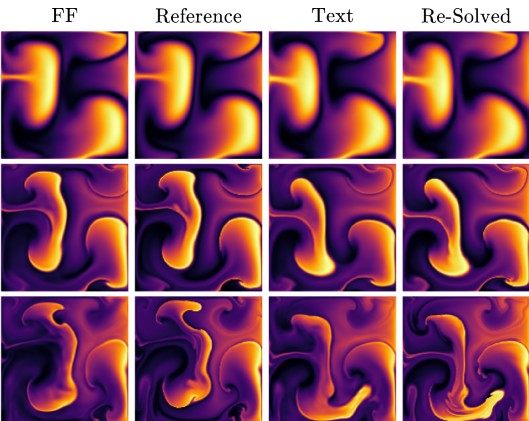

FF  Reference  Text  Re-Solved

**Table 2: Buoyancy-Driven Flow.** *Left:* Model parameters, flops, and validation losses. Text-conditioned losses* are evaluated after re-solving the ground truth. *Right:* Sample rollout; variations in generated initial states of text models can accumulate large deviations, as such, they are evaluated against a re-solved trajectory.

16. The smoke density is modeled by a passive transport equation, and as a result, the physical variables are velocity and density. Lastly, for the smoke buoyancy problem, it is more challenging to describe an initial condition with text. In particular, plumes may exhibit different shapes, locations, and sizes, in addition to the initial velocity field being even less visually distinct. Nevertheless, we investigate captioning the initial condition using a multimodal LLM. We design a prompt and provide the density field as an image; additionally, we outline individual smoke plumes in the initial density field using a Canny edge detector (Canny, 1986) and provide this segmented image in the LLM prompt as well. For additional details and examples, we refer readers to Appendix D.2.

**Results**   We consider benchmarking against the FNO, Unet, and Dilated Resnet architectures (Li et al., 2021; Ronneberger et al., 2015; Gupta & Brandstetter, 2022; Stachenfeld et al., 2022). Additionally, we benchmark against an autoregressive diffusion model (ACDM) (Kohl et al., 2024), to evaluate the need for a latent space and full spatio-temporal sample generation. To ensure a fair comparison, we provide these baselines with the initial solution and autoregressively predict the next 47 frames. We compare this to our LDM model, conditioned on the first frame (-FF) or the text prompt (-Text), and in different model sizes: Small (S), Medium (M), and Large (L), in Table 2.

Due to the underdetermined nature of text conditioning, we propose an additional evaluation metric. This is motivated by the observation that text-conditioned diffusion models often generate a variety of initial conditions that semantically satisfy a given prompt, yet this can affect the temporal rollout significantly. Although the sampled solution may not match a specific solution in the validation set, it still fulfills the semantic goal of text conditioning. Therefore, we sample a solution for each text prompt in the validation set, then use the sampled initial condition to produce a ground-truth using PhiFlow (Holl & Thuerey, 2024), which would evaluate the physical consistency of generated solutions without restricting the diversity of the LDM simulator. The losses using this re-solved trajectory are reported in Table 2, with additional visualizations in Appendix A.

We observe that LDM models conditioned on the first frame of a PDE solution can outperform current models and are an efficient class of neural solvers. Additionally, density errors are around 1.5 times the velocity errors. Furthermore, text-conditioned diffusion models are able to model a set of phenomena that both semantically satisfy the prompt and are physically consistent, although the samples may not match the validation set. In addition, many text-conditioned samples are not contained in the training or validation sets, indicating that the generative model has learned the underlying physics to generate new samples based on a text prompt. Furthermore, for this benchmark, the initial conditions are generated from a Gaussian distribution, which is challenging to describe with a prompt and real-world problems often contain more structure that natural language can better describe. Lastly, the LDM model displays good scaling behavior, with efficient training at large parameter counts.

| Model | Params | Tflops | L2 Loss | $D_{TKE}$ |
|-------|--------|--------|---------|-----------|
| FNO | 1.02B | 1.62 | 0.862 | 6.524 |
| FactFormer | 41.4M | 53.8 | 0.795 | 6.022 |
| Dil-Resnet | 24.8M | 249 | 0.707 | 6.153 |
| LDM-FF | 2.72B | 9.78 | **0.602** | **5.630** |
| LDM-Text | 2.73B | 9.43 | 0.693 | 5.653 |

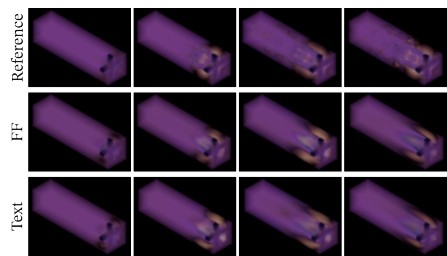

**Table 3: 3D Turbulence.** *Left:* Model parameters, flops, validation losses and log-TKE distance. *Right:* Sample rollouts for first frame and text conditioned models compared to the reference solution.

## 4.3 3D TURBULENCE

**Dataset**   To investigate scaling behavior to larger, more complex systems, we consider evaluating the LDM framework on a 3D turbulence problem of air flows over various geometric objects such as squares, elbows, U-shapes, etc. (Lienen et al., 2024). We use 36 training samples and 9 validation samples, downsampled to a resolution of $96 \times 24 \times 24$ and 1000 timesteps; however, we set the prediction horizon to 48 timesteps during evaluation. The average Reynolds number is $2e5$ and the physical variables are velocity and pressure. Additionally, the simulations are manually captioned based on the geometry of the initial condition; we include additional details in Appendix D.3.

**Results**   We consider benchmarking against FNO3D (Li et al., 2021) as well as FactFormer (Li et al., 2023b), a scalable attention-based neural solver, and Dil-Resnet (Stachenfeld et al., 2022), an accurate yet computationally expensive CNN solver. In addition to relative L2 loss, we report the L2 distance between the log turbulent kinetic energy $D_{TKE}$ between predicted and true samples, since chaotic and small-scale fluctuations in turbulence make it unlikely for similar samples to be close under Euclidean distance when evolved over time (Lienen et al., 2024). It is currently challenging for any neural surrogate to accurately model 3D turbulence, however the proposed model is able to do so more accurately than baselines and remains efficient. Although first-frame conditioning is more accurate, text conditioning is still an accurate conditioning modality. Lastly, there are significant smoothing effects which seems to be the cost for temporal stability and efficient training in 3D.

## 4.4 DISCUSSION

We show that latent diffusion models can be a viable approach to generating spatio-temporal physics simulations. Specifically, by sampling spatio-temporal noise and leveraging the ability of diffusion models to approximate complex distributions, an entire trajectory can be generated from an initial condition. This circumvents the traditional paradigm whereby solvers need to evolve solutions based on a current timestep, and aids in increasing accuracy by preventing error accumulation. We demonstrate this by plotting the time-dependent prediction error for various models in Figure 3. Additionally, the use of a latent space allows this computation to be performed efficiently and with unstructured data, and the intro-

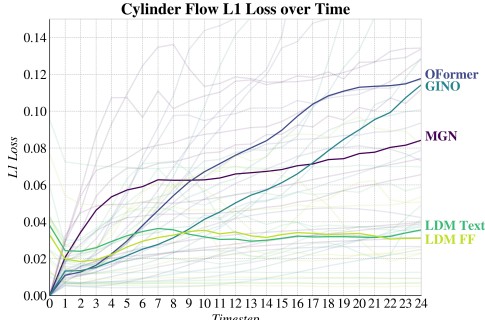

**Figure 3:** Losses at each timestep are evaluated for 10 samples. Average losses at each timestep are bolded, individual sample losses are opaque.

duction of a mesh autoencoder allows latents to be decoded where solutions need to be refined (i.e. wake regions) and onto arbitrary geometries. Lastly, the scaling behavior of the proposed model is promising and reflects similar observations of scaling diffusion and attention-based models in other domains (Peebles & Xie, 2023; Kaplan et al., 2020). The benchmarked datasets are relatively small (<5000) samples; with a larger, more diverse dataset, this scaling ability could unlock unified physics surrogates across a wide variety of phenomena.

Additionally, the introduction of text-conditioned physics simulators is an interesting and novel research direction. In constrained PDE formulations, text is a compact and accurate conditioning

modality, nearly matching first-frame conditioning in cylinder flow problems despite conditioning on around $1000\times$ less data ($\sim$1Mb vs. $\sim$1Kb), and with the added advantage of being more interpretable and usable. The diversity and ease of text-conditioned physics simulators could be a useful tool for engineers and designers to quickly test different ideas without needing to define geometries, discretizations, or generate an initial solution. After deciding on a design, the same architecture can then be used with first-frame conditioning to precisely model the resulting physical phenomena.

## 5 RELATED WORKS

### 5.1 VIDEO DIFFUSION MODELS

Many prior works have considered using diffusion models to generate videos. Initial works have focused on adapting image diffusion models to the video domain, proposing mechanisms to ensure temporal consistency or upsample spatio-temporal resolutions (Singer et al., 2022; Wu et al., 2023; Blattmann et al., 2023b). Many approaches have investigated directly generating videos by modifying the diffusion backbone (Ho et al., 2022b), with subsequent works improving the resolution, quality, and consistency of generated videos (Ho et al., 2022a; Bar-Tal et al., 2024). Within these two paradigms, an abundance of research has examined the scaling of video diffusion models (Blattmann et al., 2023a), improved conditioning methods (Girdhar et al., 2023), training-free adaptation (Khachatryan et al., 2023; Zhang et al., 2023), and video editing (Esser et al., 2023).

### 5.2 DIFFUSION MODELS FOR PDES

Previous works on adapting diffusion models for PDEs have worked with generating individual timesteps and without the use of a latent space (Kohl et al., 2024; Lienen et al., 2024; Huang et al., 2024; Yang & Sommer, 2023). To improve temporal resolution and consistency, the use of a temporal interpolator has also been investigated (Cachay et al., 2023). Although not directly used to generate PDE solutions, diffusion has also been investigated to refine PDE solutions (Lippe et al., 2023; Serrano et al., 2024). Previous work has also considered using diffusion to accomplish alternate objectives, such as unconditional generation, upsampling PDE solutions, in-painting partial PDE solutions, or recovering solutions from sparse observations (Du et al., 2024; Shu et al., 2023)

## 6 CONCLUSION

**Summary**  We have demonstrated that latent diffusion models represent a powerful class of physics simulators. Specifically, we have developed a framework for encoding and decoding arbitrary PDE data into a latent space to generate unstructured physics solutions. Furthermore, we have shown that generating full spatio-temporal solutions can mitigate error accumulation to improve accuracy. Lastly, we introduce conditioning on physics data as well as language, which is a novel modality that can be more compact, interpretable, and accessible.

**Limitations**  A limitation of diffusion models is the increased time and computation required for inference. We seek to address this using a DDIM sampler, and show that by using fewer denoising steps, diffusion models can be faster than conventional, deterministic neural solvers while maintaining accuracy in Appendix C.3. For transparency, we also compare the inference times of using DDPM sampling on the Cylinder Flow and Smoke Buoyancy problems in Appendix G. Future work could expand on this to further accelerate sampling and improve sample quality, such as distillation techniques (Salimans & Ho, 2022; Luhman & Luhman, 2021; Meng et al., 2023) and consistency models (Song et al., 2023). An additional limitation is that the LLM captioning can sometimes hallucinate, which degrades the quality of the smoke buoyancy captions and the resulting text2PDE model. Lastly, diffusing an entire rollout at once fixes the temporal resolution of the generated solution, whereas autoregressive models can generate an arbitrarily long sequence. We present preliminary work to address this limitation by using the proposed LDM model autoregressively, to alleviate the need to scale the model when predicting longer time horizons, shown in Appendix C.4.

ACKNOWLEDGMENTS

This work was funded by Fluor Marine Propulsion, LLC under purchase order number 142474. Additionally, this research used resources of the National Energy Research Scientific Computing Center (NERSC), a Department of Energy Office of Science User Facility using a Generative AI for Science NERSC award DDR-ERCAP-m4732 for 2024.

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

# A   ADDITIONAL RESULTS

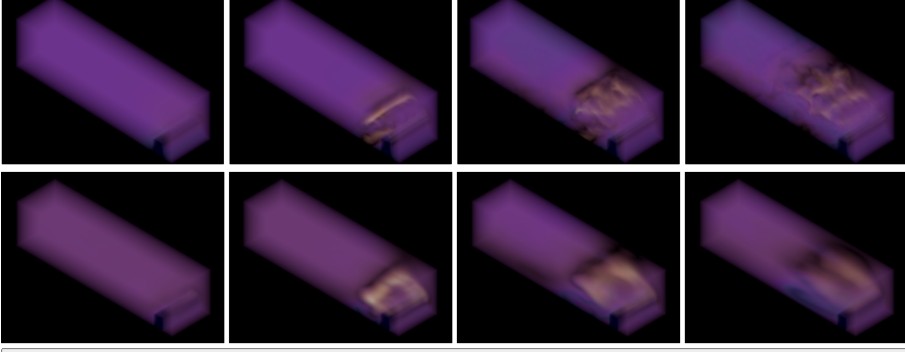

"Air flows over an obstacle resembling a low step at the bottom half, spanning the width of the domain and over a quarter the height of the domain. The step is 24 units wide and 7 units high and is located at at the bottom of the domain, starting from y = 0 to y = 6 and x = 0 to x = 23. The flow is turbulent."

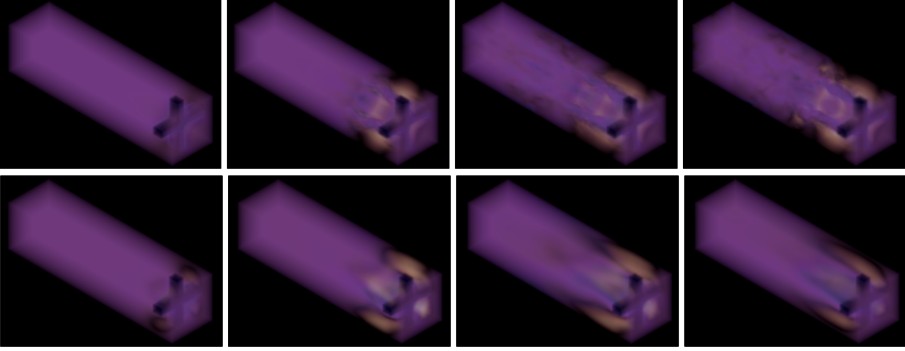

"Air flows over an obstacle resembling a narrow cross shape that consists of a narrow vertical bar that is joined together with two narrow horizontal pillars at the center left and center right. The bar is 5 units wide and 24 units high and is located at at the bottom center of the domain, starting from y = 0 to y = 23 and x = 10 to x = 14. The first pillar is 10 units wide and 5 units high and is located at at the center left of the domain, starting from y = 10 to y = 14 and x = 0 to x = 9. The second pillar is 9 units wide  and 5 units high and is located at at the center right of the domain, starting from y = 10 to y = 14 and x = 15 to x = 23. The flow is turbulent."

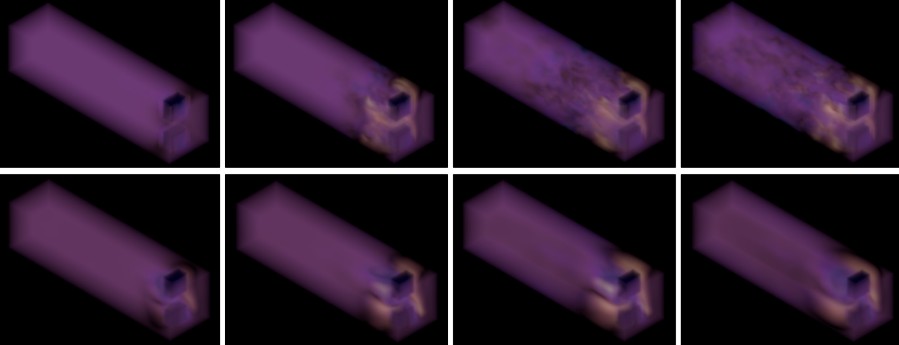

"Air flows over an obstacle resembling two wide rectangles that each span over a third of the height of the domain. The two rectangles are nearly equal in size to each other and are vertically opposite. The first rectangle is 10 units wide and 8 units high and is located at at the bottom center of the domain, starting from y = 0 to y = 7 and x = 7 to x = 16. The second rectangle is 10 units wide and 10 units high and is located at at the top center of the domain, starting from y = 14 to y = 23 and x = 7 to x = 16. The flow is turbulent."

**Figure 4:** Additional examples of text-conditioned generation of 3D turbulence samples with the velocity magnitude rendered. The true solution is shown on top, followed by the sampled solution on the bottom. While the generated solutions smoothen high-frequency features, the solutions remain stable and are broadly accurate. Rendered with vAPE4D (Koehler et al., 2024).

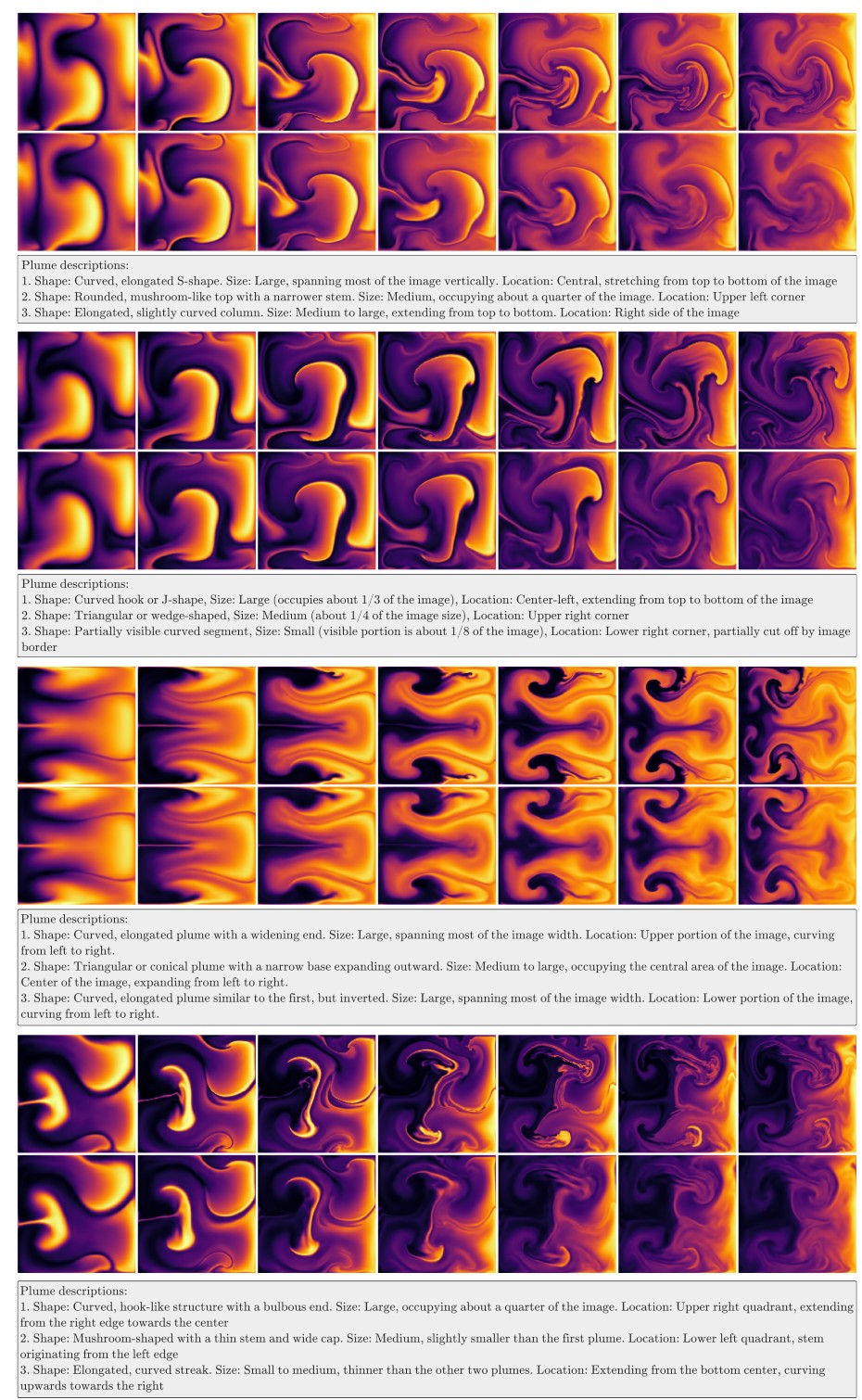

**Figure 5:** Additional examples of text-conditioned generation of smoke buoyancy examples. The true solution is shown on top, followed by the sampled solution on the bottom. The model is only given a text description of the initial frame, which is also displayed for each example with the additional observations omitted.

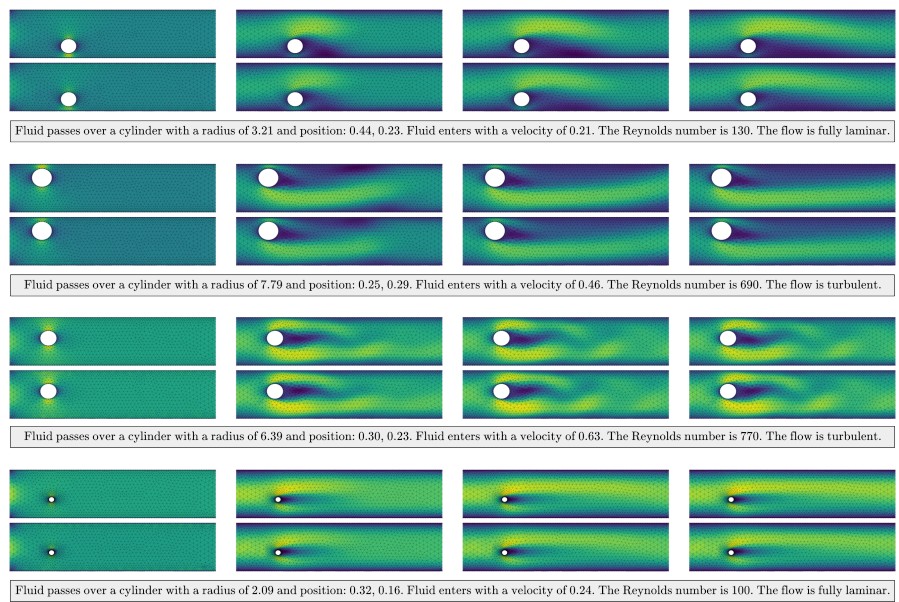

**Figure 6:** Additional examples of text-conditioned generation of cylinder flow examples. The true solution is shown on top, followed by the sampled solution on the bottom.

# B  AUTOENCODER DETAILS

| Model | L1 Loss |
|-------|---------|
| Base  | .0340   |
| +GAN  | .0311   |
| +LPIPS| .0208   |

**(a)** GAN/LPIPS Ablation for Cylinder Flow

| Model | L1 Loss |
|-------|---------|
| Base  | .00535  |
| +GAN  | .00916  |
| +LPIPS| .00938  |

**(b)** GAN/LPIPS Ablation for Smoke Buoyancy

| $\lambda_{KL}$ | KL Loss | L1 Loss |
|------|---------|---------|
| 5e-6 | 1.8e4   | 0.378   |
| 2e-7 | 4.8e4   | 0.239   |

**(c)** KL Weight ($\lambda_{KL}$) Ablation for Smoke Buoyancy

**Table 4:** Validation losses between reconstructed and true samples with various CNN backbone parameters.

## B.1  ABLATION STUDIES

We present additional results investigating the use of adversarial training and perceptual loss metrics (LPIPS) on autoencoder performance. An important consideration is that current pretrained models needed for LPIPS operate only on regularly spaced data; therefore, to maintain a fair comparison, the cylinder flow data are resampled onto a 128x32 grid to be able to evaluate an LPIPS loss. Since the benefits of using these additional methods are not clear, they are omitted from autoencoder training in the main results; however, this could be a direction for future work to improve the accuracy of LDMs. Results for the cylinder flow and smoke buoyancy problems are given in Tables 4a and 4b.

**Adversarial Training**  A common technique to improve the reconstruction accuracy of an autoencoder is to introduce a discriminator to assess whether the reconstructed input is fake or real (Esser et al., 2021). Since our proposed mesh encoder is agnostic to the input discretization, it can also be used as a discriminator by encoding a given PDE sample, flattening the latent space, and averaging the logits. In practice, the autoencoder is trained in two stages: the first stage minimizes the reconstruction error and maximizes the probability of an erroneous discriminator classification, and the second stage optimizes the discriminator when given a real and reconstructed sample.

On the cylinder flow dataset, the use of a discriminator improves autoencoder performance, however, for the smoke buoyancy dataset it does not. We hypothesize that more complex PDEs are more

challenging to learn if they are fake or real, since deviations from real samples could be reasonably interpreted as eddies or high-frequency features in more turbulent regimes. We anticipate this could be improved with additional tuning, but due to its difficulty we omit the use of adversarial training for the main results, and indeed lower losses can also be obtained but simply scaling the autoencoder and training it for longer.

**Perceptual Loss**  Common L2 and L1 losses capture global statistics on the similarity of reconstructed and true samples, however, they can misrepresent the semantic similarity of different samples. To improve these analytical losses, Zhang et al. (2018) propose to use differences in hidden activations of pretrained models as a loss metric. This is commonly implemented using a pretrained VGG16 model on ImageNet, however, this is hardly appropriate for PDE applications. We leverage a pretrained operator transformer from Hao et al. (2024), which has seen fluid dynamics data to evaluate an LPIPS loss. A final consideration is the fixed time window of pretrained PDE models; to adapt this to arbitrarily long spatio-temporal samples we autoregressively apply DPOT and average its hidden activations along a temporal rollout of the true and reconstructed samples. Again, we find that LPIPS is more beneficial for simpler physics.

**KL Penalty**  We evaluate the effect of different KL penalties on the variance of the autoencoder latent space as well as the prediction L1 loss after downstream training of a latent diffusion model in Table 4c. We find that larger KL penalties lead to lower KL losses, indicating that the latent space is closer to a Gaussian. However, this reduces the reconstruction accuracy of the autoencoder, which degrades the performance of the latent diffusion model. Alternatively, lower KL penalties result in higher-variance latent spaces, but this is outweighed by the better reconstruction capabilities.

| Ball Radius | L1 Loss |
|---|---|
| 0.045 | .0138 |
| 0.033 | .0141 |
| 0.020 | .0148 |

(a) Ball Radius Ablation for Cylinder Flow

| Model | L1 Loss |
|---|---|
| GNN | .382 |
| Neural Field | .171 |
| Ours | .011 |

(b) Architecture Ablation for Cylinder Flow

**Table 5:** Validation losses between reconstructed and true samples with various mesh learning parameters.

**Ball Radius**  We consider training the mesh encoder/decoder with different ball radii to measure the effect of the size of receptive field of the learnable aggregation and interpolation. Using the same hyperparameters for the CNN backbone, we train the mesh autoencoder at different ball radii on the cylinder flow benchmark and report the validation L1 reconstruction losses in Table 5a. We find that increasing the radius improves the reconstruction accuracy, likely since there are more points within the ball used to construct the latent grid, as well as more latent grid points used to query output points. However, this comes at an additional computational cost due to the need to integrate over more points when approximating the kernel integral.

## B.2  GNN/Neural Field Comparisons

We consider using GNN- and neural field-based approaches to encode and decode mesh data. The results for the proposed architectures on the cylinder flow problem are given in Table 5b. In general, the GNN and neural field methods do not seem to perform well. We hypothesize that this could because compressing large spatio-temporal samples is very challenging without proper inductive biases. Specifically, GNNs and attention-based neural field architectures are very general in their pooling and unpooling operations, which makes it challenging to extract compressed features and reconstruct dense outputs.

**Graph Neural Networks**  To compress and expand mesh-based data, we can apply graph pooling and unpooling methods. In particular, we consider a method of pooling and unpooling nodes based on bi-stride pooling (Cao et al., 2023). Nodes are pooled on every other BFS (Breadth-First Search) frontier in a manner that preserves connections between pooled and unpooled nodes, and doesn't require learnable parameters. Pooled and unpooled nodes at different compression levels can be

**Figure 7: Kernel Aggregation.** Neighbors to a latent grid points are used to compute an operator mapping between the input mesh and latent grid. This can also be extended to arbitrary dimensions, such as including a time dimension or extra space dimension. Additionally, this can be reversed to compute an operator mapping between the decoded latent grid and output mesh.

cached and used for the expansion or contraction of graphs. At the encoder, pooled nodes use message passing between unpooled nodes in a defined neighborhood to aggregate information and downsample the graph. At the lowest resolution, we apply cross-attention between the node list and a fixed latent vector, similar to Perceiver pooling (Jaegle et al., 2021). To interpolate a fixed latent back onto a graph, a we use cross-attention between a fixed latent and a learnable latent node list, which is truncated to the correct dimension. The low-resolution graph can be used as supernodes in a cached upsampled graph to pass messages to their unpooled neighbors until the original graph is reconstructed.

**Neural Fields** Neural fields represent a mapping between a set of coordinates and their functional values. While this is restricted to modeling a single sample, the addition of a condition in conditional neural fields allows this framework to be adapted to different PDE samples. To compress an arbitrary spatio-temporal vector, the spatial and temporal axes are separately aggregated using Perceiver pooling. Specifically, the architecture alternates between computing the cross-attention on spatial or temporal points and a fixed vector. This can then be downsampled or upsampled using conventional CNN methods. To interpolate a fixed latent back onto an output mesh, the fixed latent can be seen as a condition for a neural field queried at the output mesh. We implement this based on the CViT architecture (Wang et al., 2024). A transformer encoder computes self-attention to encode the latent vector into a condition, while the query coordinates are interpolated from a learned latent grid. A transformer decoder then calculates the cross-attention between the query points and the condition, which is decoded to the reconstruction.

### B.3 KERNEL AGGREGATION/INTERPOLATION

Given an input that is arbitrarily discretized $\mathbf{u}(t, \mathbf{x}_m)$, it can be seen as the set of values in which the true function $\mathbf{u}$ is given at. We seek to map this function $\mathbf{u}(y)$ to another function $\mathbf{q}(x)$, where we can control the discretization to be uniform. From Li et al. (2020), this mapping can be approximated by a learnable kernel integral, specifically $\mathbf{q}(x) = \int_\Omega \kappa_\theta(x, y)\mathbf{u}(y)dy$, where $\mathbf{q}$ is given on coordinates $x$ and $\mathbf{u}$ is given on coordinates $y$. Importantly, we can choose the query coordinates $x$ and evaluate this kernel integral at these arbitrary queries; however, choosing uniform query points $x$ is convenient for the CNN backbone. In practice, this is implemented as in Figure 7. Nodal values consisting of pressure and velocity are given on a mesh, and a uniform latent grid is defined. To make the kernel integral tractable, the integration domain is truncated to a local ball, where neighbors within a certain radius of latent grid points contribute to the operator mapping. Again, from Li et al. (2020), this local Monte Carlo approximation can still be accurate as the error scales with $m^{-1/2}$, where $m$ is the number of points sampled.

We can reverse this process at the decoder to interpolate the decoded latent grid onto the output mesh. Specifically, the latent function $\mathbf{q}_d$ is now mapped to the output function $\mathbf{u}_d$ by the kernel integral $\mathbf{u}_d(y) = \int_\Omega \kappa_\theta(y, x)\mathbf{q}_d(x)dx$. Importantly, the operator mapping ensures that the output mesh can be queried at arbitrary locations $y$, rather than fixing the decoded solution (both for the autoencoder and LDM) to a single discretization. This discretization invariance can be seen when using the LDM to generate a solution based on a text prompt, but decoding it onto arbitrary query points, such as in Figure 8, and is empirically reproduced by Li et al. (2023c) as well.

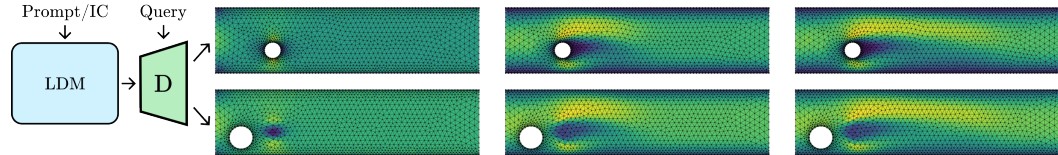

**Figure 8: Discretization Invariance.** Given a sampled latent solution from the conditional LDM, it can be decoded onto an arbitrary set of query points.

## C  DIFFUSION DETAILS

### C.1  DENOISING ABLATIONS

| Model | $L_{simple}$ |
|---|---|
| Noise | 0.056 |
| Velocity | 0.072 |

**(a)** Parameterization

| Model | $L_{simple}$ |
|---|---|
| Fixed $\Sigma$ | 0.056 |
| Learned $\Sigma_\theta$ | 0.059 |

**(b)** Learned $\Sigma_\theta$

| Model | $L_{simple}$ |
|---|---|
| Scale=1.0 | 0.034 |
| Scale=0.2 | 0.015 |

**(c)** Latent Space Scaling

| Model | $L_{simple}$ |
|---|---|
| Linear | 0.034 |
| Cosine | 0.069 |

**(d)** Noise Scheduler

**Table 6:** Validation reweighted VLB losses evaluated for different settings of the denoising process

**Denoising Parameterization**   We investigate parameterizing the reverse process by predicting the velocity $v = \sqrt{\bar{\alpha}_n}\epsilon - \sqrt{1 - \bar{\alpha}_n}\mathbf{x}_0$, rather than the noise $\epsilon$ (Salimans & Ho, 2022). This is motivated by the idea that the prediction of pure noise becomes increasingly unstable as the number of timesteps decreases. We find that v-prediction slightly increases the VLB loss, however, this effect is not significant.

**Learning $\Sigma_\theta$**   Following Nichol & Dhariwal (2021) we investigate parameterizing the reverse process variance as $\Sigma_\theta(z_t, t) = \exp(v \log \beta_t + (1-v) \log \tilde{\beta}_t)$, where $v$ is a learned vector that modulates the transition between $\beta_t$ and $\tilde{\beta}_t$. Furthermore, the loss is modified to minimize the log-likelihood. We find that learning $\Sigma_\theta$ does not significantly effect the reverse process.

**Latent Space Scaling**   Following Rombach et al. (2022), we scale the latent space to approximate unit variance across latent encodings of the training set. When the latents are too large, the addition of Gaussian noise with unit variance does not sufficiently noise the latents ($z_T \neq \mathcal{N}(0, I)$), which harms the denoising process during inference. While the KL-loss could be increased during autoencoder training to limit the variance of the learned latent space, this comes at the cost of reduced reconstruction accuracy. As such, scaling the latent space is an important consideration, and a proper scaling factor is needed to achieve good results.

**Noise Scheduler**   Following Nichol & Dhariwal (2021), we evaluate using a cosine or linear noise scheduler. We find that empirically, a linear noise scheduler performs better in our case; however, optimizing the noise schedule is still an area for future work.

### C.2  DIFFUSION BACKBONE ARCHITECTURES

We evaluate parameterizing the reverse process using a 3D Unet or DiT in Table 8. In both cases, the models need to predict spatio-temporal noise, either by using 3D convolutions or patchifying across space and time. Models are trained with approximately the same parameter count and conditioning on either the first frame or a text prompt for the cylinder flow problem. We report L1 losses between generated samples and the validation set. We find that a DiT backbone slightly outperforms a Unet backbone.

| Model | Val L1 Loss |
|---|---|
| Unet-FF | 0.0416 |
| Unet-Text | 0.0492 |
| DiT-FF | 0.0385 |
| DiT-Text | 0.0404 |

**Table 8:** Different model architectures for the Cylinder Flow problem

| Model | S | Inf. Time | Rel. L2 Loss |
|---|---|---|---|
| GINO | - | 0.112 | 0.2445 |
| MGN | - | 1.683 | 0.2617 |
| OFormer | - | 1.336 | 0.3386 |
| LDM$_S$-FF | 10 | 0.4572 | 0.1595 |
| | 20 | 0.6119 | 0.1583 |
| | 50 | 1.0844 | 0.1579 |
| | 100 | 1.8716 | 0.1595 |
| | 1000* | 15.753 | 0.1522 |
| LDM$_M$-FF | 10 | 0.5672 | 0.1313 |
| | 20 | 0.8654 | 0.1267 |
| | 50 | 1.7188 | 0.1306 |
| | 100 | 3.1561 | 0.1281 |
| | 1000* | 27.895 | 0.1309 |
| LDM$_S$-Text | 10 | 0.4711 | 0.1941 |
| | 20 | 0.6405 | 0.1980 |
| | 50 | 1.1522 | 0.1949 |
| | 100 | 1.9936 | 0.1903 |
| | 1000* | 16.251 | 0.1796 |
| LDM$_M$-Text | 10 | 0.5770 | 0.1474 |
| | 20 | 0.8836 | 0.1536 |
| | 50 | 1.7635 | 0.1513 |
| | 100 | 3.2456 | 0.1495 |
| | 1000* | 29.096 | 0.1476 |

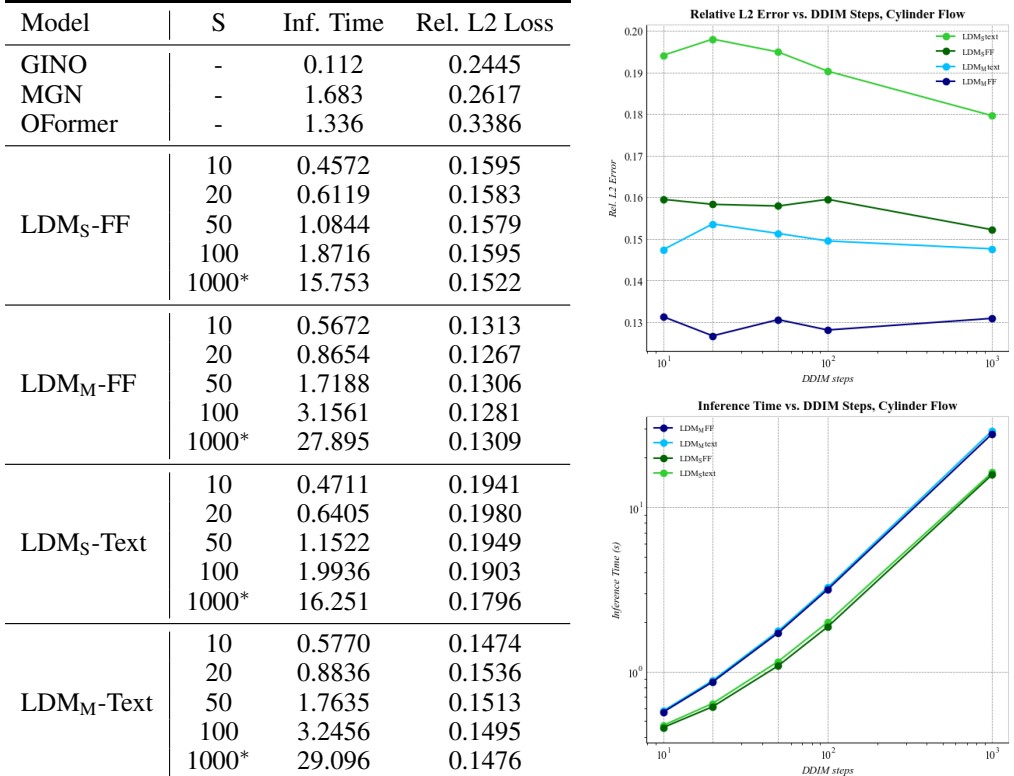

**Table 7: Cylinder Flow Sampling.** Comparison of inference times (s) and relative L2 errors of different baselines and LDM models under varying numbers of denoising steps S using DDIM or DDPM* sampling.

## C.3 DDIM SAMPLING

We evaluate the use of DDIM to accelerate sampling after training an LDM with the denoising objective. We implement a DDIM sampler (Song et al., 2022) and evaluate the inference time and relative L2 loss of sampled solutions on the Cylinder Flow problem at different denoising steps S. These results are presented in Table 7 on a single NVIDIA A100 GPU. We find that DDIM can recover most of the solution even at a small number of denoising steps, and in certain cases produce higher quality samples. Through the use of DDIM, in combination with using a latent space and spatio-temporal sampling, diffusion models can produce solutions faster than conventional, deterministic neural solvers in addition to being more accurate. We hypothesize that DDIM is effective due to fundamental differences in the modeled distributions within the image and PDE domains.

We consider the underlying conditional distribution $p_\theta(\mathbf{u}|\mathbf{c})$ that is approximated by conditional diffusion. Within image domains, this distribution has many modes in order to cover a diverse set of sampled images for a given prompt, however, in PDE settings this distribution ideally only has a single mode since when conditioned on a past timestep, there should only be a single sampled solution. Therefore, in PDE settings, we can expect Langevin sampling to converge quickly, since the distribution that is sampled from has only a single mode. Furthermore, since score estimates are still trained using the original denoising objective, they are still accurate at the initial sample $\mathbf{x}_0$; therefore, although Langevin sampling can start far from $p_\theta(\mathbf{u}|\mathbf{c})$, the score estimates are still accurate enough to quickly approach the desired mode.

One method of empirically evaluating this could be to calculate the Jacobian of the estimated score function $\nabla_x s_\theta(x)$ for diffusion models trained in PDE and image settings. The Jacobian of the score function would correspond to the Hessian of the estimated underlying potential function $\nabla_x^2 F_\theta(x)$, which is related to the mean mixing time of the Langevin sampler. The mean mixing time is correlated with the inverse of the Hessian determinant (Bovier et al., 2004), which could measure the complexity of the estimated distribution $p_\theta(x) \propto F_\theta(x)$ and its effect on sampling times.

| Model | Rel. L2 Loss |
|---|---|
| GINO | Unstable |
| MGN | 0.171 |
| OFormer | 0.088 |
| $LDM_{AR}$-FF | 0.068 |
| $LDM_{AR}$-Text | 0.085 |

**Table 9:** *Left:* Autoregressive $LDM_M$ results on Cylinder Flow, using identical hyperparameters to Table 1. *Right:* Framework for autoregressively generating text-conditioned PDE samples.

### C.4 AUTOREGRESSIVE LDMs

Although sampling complete PDE trajectories can result in lower error accumulation, this fixes the temporal horizon and would require scaling the model in order to achieve longer rollouts. To address this, we present results on using the proposed LDM framework autoregressively, albeit with a large prediction window. For LDM models conditioned on physical field values, this can be easily done by passing the last frame of the predicted solution back to the LDM model. However, for text-conditioned LDM models, the prompt is only relevant for the first prediction. To address this, we investigate a conditioning strategy in which the model can dynamically switch between text- and physics-based conditioning during inference by padding the unused modality with zeros, shown in Table 9. From a user perspective, querying the model with a text prompt remains the same; however, the model internally uses different conditioning modalities at different steps.

To validate this, we train the LDM model on the Cylinder Flow problem to predict 100 timesteps, in four windows of 25 timesteps each. We also compare this approach with neural solver baselines re-trained with a longer prediction horizon. To maintain a comparison to the original experiment, we keep the hyperparameters and model sizes the same and present the results in Table 9 for the medium-sized LDM model. Predictably, errors grow across all models in the longer time horizon setting, with GINO becoming unstable due to error propagation. Notably, the text-conditioned LDM model incurs a performance loss presumably due to switching modalities during inference.

## D CONDITIONING DETAILS

### D.1 CYLINDER FLOW CAPTIONING

For the cylinder flow problem, we extract the cylinder radius, position, inlet velocity, and Reynolds number from the first frame. Since the cylinder radius is much smaller than the domain, it is given in centimeters while the position is given in meters. Additionally, we determine the flow regime based on the Reynolds number: Re<200 = laminar, Re<350 = transition, Re>350 = turbulent. One limitation is that this is overly simplistic; flow regimes are generally problem-specific and can vary even with the same Reynolds number. In addition, there are varying standards to characterize these flow regimes from Reynolds numbers (Blevins, 1990; Zdravkovich, 1990; Stringer et al., 2014a). However, given the lack of resources to manually label this dataset, we adopt this simple strategy since these semantic differences do not greatly affect performance. Values are inserted in the prompt:

```
"Fluid passes over a cylinder with a radius of {cylinder_radius:.2f} and position: {cylinder_pos[0]:.2f},
{cylinder_pos[1]:.2f}. Fluid enters with a velocity of {inlet_velocity:.2f}. The Reynolds number is
{reynolds_number}. The flow is {flow_regime}."
```

Since the text prompt is procedurally generated, we ablate the full prompt against two baselines. The first baseline considers generating a text prompt solely with the extracted values, without any of the surrounding text, and uses the same RoBERTa model to encode the condition (Text-Ablate). The second baseline embeds the extracted values, normalized between -1 and 1, into a vector and uses an MLP to encode the condition (Vector-Ablate). We train these two baselines and a model using the full prompt and report the validation L1 loss in Table 10. We find that the extra

| Model | Val L1 Loss |
|---|---|
| Full Prompt | 0.0404 |
| Text-Ablate | 0.2168 |
| Vector-Ablate | 0.1029 |

**Table 10:** Different model architectures for the Cylinder Flow problem

context is beneficial for language models in understanding the meaning of different extracted values. Furthermore, using a language model can be a more expressive way of encoding low-dimensional physical values into a meaningful condition.

## D.2 SMOKE BUOYANCY CAPTIONING

The captioning procedure for the smoke buoyancy dataset is more challenging, since the initial density and velocity fields do not contain features that are easily extracted and described by text. To address this, we prompt an LLM (Claude 3.5 Sonnet) to describe an image of the initial density field. To improve the quality of generated captions, we also use a Canny edge detector to outline shapes in the image and provide this segmented image in the LLM prompt. The prompt is given below.

```
You will be analyzing an image of a simulation of the Navier-Stokes equations to identify smoke plumes,
count them, and describe their characteristics. You will also be provided an image with the boundaries of
the smoke plumes drawn over the simulation in green.

Follow these instructions carefully:

First, examine the provided image:
<image>
Second, examine the provided segmented image:
<segmented_image>

To identify plumes:
1. Look for regions of bright color, which contrast against the darker background.
2. Pay attention to color variations and changes in the shape of the plumes.
3. Use the green lines in the segmented image to help identify the boundaries of the plumes.

To count the number of plumes:
1. Scan the entire image systematically.
2. Use the green lines in the segmented image to help separate different plumes.
3. Pay attention to darker regions that separate different plumes.

To describe each plume's shape, size, and location:
1. Shape: Characterize the overall form (e.g., column-like, mushroom-shaped, dispersed cloud,
triangular, curved) and changes in shape of the plume.
2. Size: Estimate the relative size compared to other elements in the image (e.g., small, medium, large)
and describe how the plume changes in size.
3. Location: Describe the position using general terms (e.g., upper left corner, center, lower right)
or in relation to other smoke plumes.

Remember to use the segmented image to help describe each plume's shape, size, and location.

Present your findings in the following format:

<answer>
Number of plumes identified: [Insert number]
Plume descriptions:
1. [Shape], [Size], [Location]
2. [Shape], [Size], [Location]
(Continue for each plume identified)
Additional observations: [Only include any relevant details about the overall patterns or shapes observed.
Pay attention to symmetry and how the plumes interact with each other.]
<answer>

Remember to be as descriptive and accurate as possible in your analysis.
```

We find that using a segmented image improves quantitative and qualitative metrics in labeling physics simulations. In particular, the correct number of smoke plumes are identified more often, and the resulting descriptions are richer and more detailed. We give an example in Figure 9. Additionally, to inform models of the forcing term, the buoyancy factor is prepended to the LLM answer by inserting it into the template: "The buoyancy factor is {}.". We also compare the responses of Claude 3.5 Sonnet, Gemini 1.5 Pro, and GPT-4o in Figure 9.

## D.3 3D TURBULENCE CAPTIONING

Due to the small number of samples, we manually caption this dataset. To maintain consistency across human-generated captions, we follow a template that asks for a pre-specified set of features. In particular, we look to describe qualitative features, the position, and the size of the shape, as well as how many shapes are present. An example template is provided below; however, the templates can vary based on the quantity and composition of the shapes:

```
"Air flows over an obstacle resembling a {qualitative_label}. The {shape} is {shape_label} and is located at
{position_label}. The flow is turbulent."
```

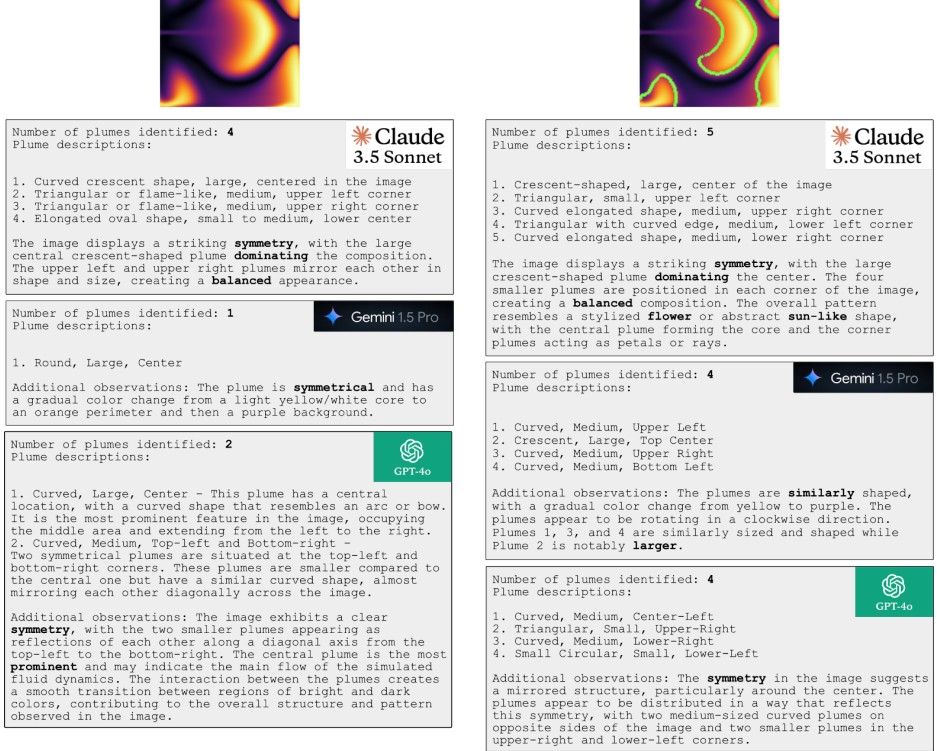

**Figure 9:** Comparison of LLM outputs after prompting with the original image, or both the original and a segmented image. We also compare responses from Claude 3.5 Sonnet, Gemini 1.5 Pro, and GPT-4o.

### D.4 CLASSIFIER-FREE GUIDANCE

Classifier-free guidance can be a method to modulate the strength of the conditioning information. This is implemented by using conditional dropout during training and combining conditional and unconditional noise estimates during inference: $\tilde{\epsilon}_\theta(\mathbf{z}_n, n, \mathbf{c}) = (1 + w)\epsilon_\theta(\mathbf{z}_n, n, \mathbf{c}) - w\epsilon_\theta(\mathbf{z}_n, n)$. The weight $w$ modulates the tradeoff between sample quality and diversity, with larger values increasing the guidance strength and decreasing sample diversity. Following Lin et al. (2024), we also rescale the conditional portion of the sample generation process with the recommended value of $\phi = 0.7$. Empirically, we find that only at very large or small weights does the generated sample change appreciably. We plot the trends in Figure 10 for first-frame conditioning. We find that at large values of $w$, much of the generated solution is saturated and many features are washed out. At low values of $w$, many features are not present and the generated solution displays a lack of phenomena. We hypothesize that classifier-free guidance is not necessary for physics applications where the primary concern is the numerical accuracy of solutions, rather than diversity or visual quality and indeed the loss of models trained without classifier-free guidance ($w = 1$) is often better than using other values of $w$.

## E DATASET DETAILS

### E.1 CYLINDER FLOW

We use data from Pfaff et al. (2021), which is generated from water flowing over a cylinder constrained in a channel, modeled by the incompressible Navier-Stokes equations:

$$\frac{\partial \mathbf{v}}{\partial t} + \mathbf{v} \cdot \nabla \mathbf{v} = \nu \nabla^2 \mathbf{v} - \frac{1}{\rho}\nabla p + \mathbf{f}, \qquad \nabla \cdot \mathbf{v} = 0 \qquad (7)$$

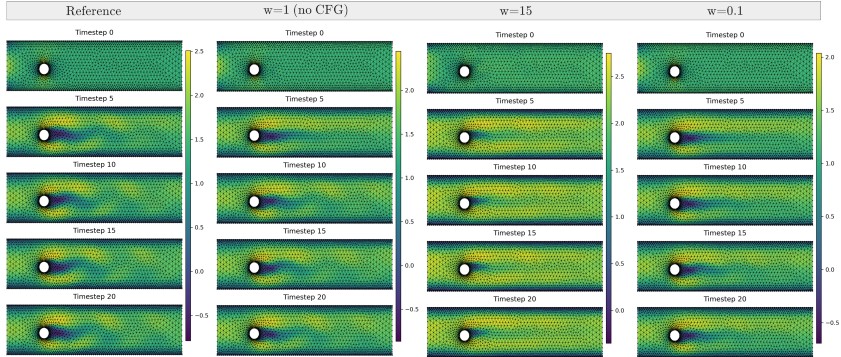

**Figure 10:** Comparison of different weights used for classifier-free guidance. In general, using vanilla conditional generation results in the most accurate solutions.

The original data are generated using COMSOL, with 600 timesteps from $t = 0$ to $t = 6$ on a spatial domain $(x = [0, 1.6], y = [0, 0.4])$, with varying inlet velocities and cylinder sizes/positions. To speed up our experiments, we downsample along the temporal dimension to 25 timesteps. We use 1000 samples for training and 100 samples for validation. Both datasets contain samples with Reynolds numbers ranging from 100-1500. Additionally, we identify Karman vortex streets in data samples by calculating the norm of the temporal velocity derivative $||\frac{dv_x}{dt} + \frac{dv_y}{dt}||_2$ and comparing it to a cutoff; if this quantity remains large or increases over time then periodic vortex shedding continues to occur and if this quantity decreases or goes to zero over time then the so-

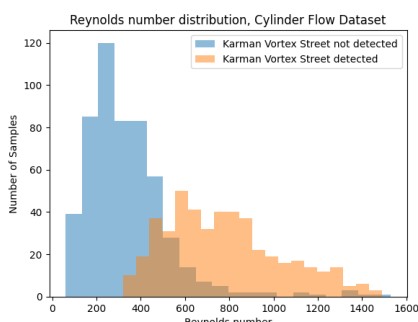

**Figure 11:** Distribution of Reynolds numbers within the training set. Samples with Karman Vortex streets are also identified.

lution has reached a laminar steady state. The Reynolds number of training data samples, as well as Karman vortex street behavior is plotted in Figure 11. The observed Karman vortex streets within this dataset are consistent with the literature describing its occurrence at Reynolds numbers between 300-2e5 (Chen, 1985; Gabbai & Benaroya, 2005).

### E.2 SMOKE BUOYANCY

We use data from Gupta & Brandstetter (2022), which is generated from smoke driven by a buoyant force constrained in a box, modeled by the incompressible Navier-Stokes equations coupled with an advection equation representing the smoke density $d$:

$$\frac{\partial \mathbf{v}}{\partial t} + \mathbf{v} \cdot \nabla \mathbf{v} = \nu \nabla^2 \mathbf{v} - \frac{1}{\rho} \nabla p + \mathbf{f}, \qquad \nabla \cdot \mathbf{v} = 0 \tag{8}$$

$$\frac{\partial d}{\partial t} + \mathbf{v} \cdot \nabla d = 0 \tag{9}$$

The original data are generated using PhiFlow (Holl & Thuerey, 2024), with 56 timesteps from $t = 18$ to $t = 102$, which results in $\Delta t = 1.5s$. Furthermore, the simulation is solved on a spatial domain $(x = [0, 32], y = [0, 32])$ with a resolution of $128 \times 128$. To ensure compatibility with downsampling operations, which occur with a factor of 2, we truncate the simulation horizon to $t = 48$. We use 2496 samples for training and 608 samples for validation.

### E.3 3D TURBULENCE

We use data from Lienen et al. (2024), which is generated from LES simulation in OpenFOAM with a viscosity of $\nu = 10^{-5}$. The original data is given at a resolution of $192 \times 48 \times 48$ and at 5000

timesteps. A variety of shapes (steps, corners, pillars, teeth, bars, elbows, donut, U-shape, T-shape, H-shape, squares, crosses, platforms, etc.) are used to model flows around different objects, and for additional visualizations we refer readers to page 18 of the original paper (Lienen et al., 2024). Details on the calculation of the log-TKE distance can also be found in the original paper.

# F    TRAINING DETAILS

To maintain a consistent setup in all experiments, the only ground truth information provided to each model is from the first frame of the simulation. Additionally, to maintain a fair FLOPs comparison, the compute required for autoregressive models during training is multiplied by the number of timesteps. This is because the LDM model considers all timesteps at once during training, while autoregressive models only consider a single timestep.

## F.1    CYLINDER FLOW

All baselines were trained on a single NVIDIA RTX 6000 Ada GPU with a learning rate of $10^{-5}$ until convergence. LDM models were trained on a single NVIDIA A100 40GB GPU. For each model, we describe the key hyperparameters.

**GINO**    We train a GINO model with a latent grid size of 64 and GNO radius of 0.05. Furthermore, the FNO backbone uses a hidden dimension of 128 and 32 modes.

**MeshGraphNet**    We train a MGN model with 8 layers and a hidden size of 1024.

**OFormer**    We train an OFormer model with 6 layers and a hidden size of 512.

**LDM**    We train an autoencoder with a latent grid size of 64, GNO radius of 0.0425, hidden dimension of 64, and 3 downsampling layers, resulting in a latent size of $16 \times 16 \times 16$, which is a compression ratio of around 48. Additionally, the DiT backbone uses 1000 denoising steps, a patch size of 2, a hidden size of (512/1024), and a depth of (24/28) depending on model size.

## F.2    SMOKE BUOYANCY

All baselines were trained on a single NVIDIA RTX 6000 Ada GPU with a learning rate of $10^{-4}$ until convergence. LDM models were trained on four NVIDIA A100 40GB GPUs. For large models, four NVIDIA A100 80GB GPUs were used to handle memory requirements. For each model, we describe the key hyperparameters.

**FNO**    We train a FNO model with a hidden dimension of 192, 24 modes, and 6 layers.

**Unet**    We train a Unet model with 4 downsampling layers and a hidden size of 128, with architecture modifications (Unet$_{mod}$) proposed in Gupta & Brandstetter (2022).

**Dil-Resnet**    We train a Resnet with a hidden dimension of 256, 4 layers, and a dilation of up to 8.

**ACDM**    We train an ACDM model with a single conditioning frame and with a Unet backbone with a hidden dimension of 256 and 4 downsampling layers.

**LDM**    We train an autoencoder with a hidden dimension of 64 and 4 downsampling layers, resulting in a latent size of $6 \times 16 \times 16$, which is a compression ratio of 512. Additionally, the DiT backbone uses 1000 denoising steps, a spatial patch size of 2 and a temporal patch size of 1, a hidden size of (512/1024/1536/2304) and a depth of (24/28) depending on model size.

## F.3    3D TURBULENCE

All baselines were trained on a single NVIDIA A100 GPU with a learning rate of $10^{-5}$ until convergence. LDM models were trained on four NVIDIA A100 80GB GPUs were used to handle memory requirements. For each model, we describe the key hyperparameters.

**FNO**    We train a FNO model with a hidden dimension of 192, 16 modes, and 6 layers.

**Factformer**   We train a Factformer model with a hidden dimension of 256, 8 heads, and 18 layers.

**Dil-Resnet**   We train a Resnet with a hidden dimension of 128, 4 layers, and a dilation of up to 8.

**LDM**   We train an autoencoder with a hidden dimension of 64 and 4 downsampling layers, resulting in a latent size of $6 \times 12 \times 3 \times 3$, which is a compression ratio of 4096. The DiT backbone uses 1000 denoising steps, a patch size of $(1, 1, 2, 2)$, a hidden size of 2048, and a depth of 28.

## G   INFERENCE TIME COMPARISON

(a) Inference Time for Cylinder Flow

| Model | Params | Time |
|---|---|---|
| GINO | 72M | 0.11 |
| MGN | 101M | 1.68 |
| OFormer | 131M | 1.34 |
| LDM$_S$-FF | 198M | 15.75 |
| LDM$_M$-FF | 667M | 27.89 |
| LDM$_S$-Text | 313M | 16.25 |
| LDM$_M$-Text | 804M | 29.10 |

(a) Inference Time for Cylinder Flow

(b) Inference Time for Smoke Buoyancy

| Model | Params | Time |
|---|---|---|
| FNO | 510M | 1.52 |
| Unet | 580M | 2.33 |
| Dil-Resnet | 33.2M | 5.83 |
| ACDM | 404M | 317.8 |
| LDM$_S$-FF | 243M | 16.86 |
| LDM$_M$-FF | 725M | 25.23 |
| LDM$_L$-FF | 1.55B | 57.44 |
| LDM$_S$-Text | 334M | 16.30 |
| LDM$_M$-Text | 825M | 26.49 |
| LDM$_L$-Text | 2.69B | 84.46 |

(b) Inference Time for Smoke Buoyancy

(c) Inference Time for 3D Turbulence

| Model | Params | Time |
|---|---|---|
| FNO | 1.02B | 1.57 |
| FactFormer | 41.4M | 5.15 |
| Dil-Resnet | 24.8M | 38.8 |
| LDM$_L$-FF | 2.72B | 75.2 |
| LDM$_L$-Text | 2.73B | 75.7 |

(c) Inference Time for 3D Turbulence

**Table 11:** Inference time to predict single temporal rollout (batch size = 1), evaluated on a single NVIDIA RTX 6000 Ada GPU or NVIDIA A100 GPU (3D Turbulence). Reported times are in seconds and are averaged over nine validation samples.

For transparency, we report inference times of all models on the Cylinder Flow, Smoke Buoyancy, and 3D Turbulence benchmarks in Table 11. We reproduce previous results that GINO is an efficient architecture for unstructured problems (Li et al., 2023c). Furthermore, we reproduce previous results demonstrating the increased compute needed for dilated Resnets (Li et al., 2023b). Lastly, we find that our latent diffusion framework is much faster than prior work on autoregressive diffusion or learning directly in the pixel space.

However, we recognize that the increased latent diffusion inference times may be a potential limitation. The current implementation uses 1000 denoising steps, which leaves ample room for accelerating inference. Indeed, using a DDIM sampler, as shown in Appendix C.3, is able to vastly accelerate inference and outperform conventional, deterministic neural solvers in both speed and accuracy.

To compare the proposed method to a numerical baseline, the numerical solver needs to be coarsened to match the accuracy of the neural solver, as well as a reasonable effort needs to be made to compare against a state of the art numerical solver (McGreivy & Hakim, 2024). The solvers used for the Cylinder Flow, Smoke Buoyancy, and 3D Turbulence benchmarks are COMSOL, PhiFlow and OpenFOAM, which are all high-performance numerical packages. We do not have access to the COMSOL code to solve the cylinder flow problem with reduced accuracy. However, the original authors report a speedup of 11-290x (Pfaff et al., 2021), which may be enough margin to overcome coarsening the simulation to 90% of its original accuracy (see Table 1, model is reported with a 13% relative L2 error). Additionally, we also do not have access to the OpenFOAM simulation, but the original authors report a simulation time of around 10-20 minutes on 16 CPU cores (Lienen et al., 2024).

For the smoke buoyancy benchmark, we have access to PhiFlow and, in fact, use it to re-solve text-conditioned samples. We compare the accuracy and speed of using PhiFlow at lower resolutions with latent diffusion models in Table 12. PhiFlow was solved faster on our CPU (AMD Ryzen Threadripper PRO 5975WX 32-Cores) than on our GPU (NVIDIA RTX 6000 Ada), so numerical solver times are reported using the CPU. When running lower-resolution PhiFlow simulations, the

original initial condition was downsampled, evolved with the solver, and upsampled using bicubic interpolation. There are tangible accuracy and speed gains from using a latent diffusion model; however, faster inference techniques such as DDIM are needed to vastly outperform numerical solvers. Also of note is that text2PDE generation is something that numerical solvers cannot do.

| Model | Inference Time (s) | Val L1 Loss |
|---|---|---|
| $LDM_S$-FF | 16.863 | 0.154 |
| $LDM_M$-FF | 25.233 | 0.139 |
| $LDM_L$-FF | 57.442 | 0.118 |
| $LDM_S$-Text | 16.294 | 0.174 |
| $LDM_M$-Text | 26.494 | 0.132 |
| PhiFlow-32 | 69.266 | 0.172 |
| PhiFlow-64 | 70.569 | 0.116 |
| PhiFlow-128 | 75.468 | 0 (Ground-Truth) |

**Table 12:** Comparison of inference speed and accuracy. PhiFlow-X denotes the spatial resolution used for numerical simulation. PhiFlow speed and accuracy values are averaged over five validation samples.

