# OpenReview forum: "Text2PDE: Latent Diffusion Models for Accessible Physics Simulation"
_ICLR.cc/2025/Conference — ICLR 2025 Poster_

### Official Review · Reviewer_zsUD · 2024-10-30

**Soundness:** 3
**Presentation:** 2
**Contribution:** 2
**Rating:** 3
**Confidence:** 5

**Summary:**

This work presents a more accessible physics simulation by introducing the language model so that the physics solver can be interacted by using the test, called text2PDE. The proposed method is verified cylinder flow and buoyancy-driven flow although they cannot demonstrate enough the claimed effect.

**Strengths:**

1. The developed framework encodes and decodes arbitrary PDE data into a latent space to generate unstructured physics solutions.
2. The language model is applied in the framework to make the PDE solver more accessible.
3. The paper applies two experiments to demonstrate the superior performance of the proposed method.

**Weaknesses:**

1. The reviewer raises doubts about the novelty and contribution of the work, seemingly only using the encoding of the language model as a condition to generate solutions of PDEs.
2. Why do the two experiments adopt different baselines and lack baselines such as FNO for cylinder flow?
3. From line 047 to 050, while this statement may seem obvious, it's not a scientific issue. It is necessary to have physics and numerical knowledge to build PDE solvers. At the same time, approaches based on FNO, DeepOnet and similar numerically-driven methods can learn accurate PDE solvers without requiring physics knowledge.
4. From line 053, this work deviates from the challenges described above, and there are concerns about potential overclaiming. Based on the author's two experiments, the proposed model only works in specific PDEs, spatiotemporal dimensions, and simple scenarios where the required physics knowledge is quite basic. In other words, the authors failed to address a crucial question: for complex PDE systems, how to represent the system using language, when mathematical formulas are clearly more suitable, while it is the main challenge the reviewer understand from the previous paragraph.
5. Spatial and temporal resolution is crucial for PDE solvers. How does this work handle different spatial resolutions? Specifically, how does this method distinguish between initial conditions with different resolutions within the same system?
6. Could you provide more details about the autoencoder in Sec. 3.1? This part is quite abstract and difficult for readers to understand intuitively. Please briefly explain the autoencoder's architecture, input/output structure, and training methodology. Additionally, during the diffusion model training, is the encoder trained simultaneously or are its parameters fixed?
7. In line 235, the decoder appears to be CNN-based, which imposes constraints on the spatial dimensions of the output solutions. Furthermore, interpolation operations can lead to inaccuracies in the results, even when solutions on regular grids are accurate.
8. In the method section, the authors have placed too many crucial details in the appendix, like Appendix C.2, B.1, B.2, D, making it impossible to understand the method without referring to the appendix.
9. The selected parameters for cylinder flow are overly simplistic, only covering laminar flow cases. Generally, the Kármán vortex street phenomenon is more common in cylinder flows, yet the authors did not include this scenario.
10. All data used is 2D, and it's unclear whether this model can handle 1D and 3D experiments. The lack of such experiments significantly reduces the persuasiveness of the work.
11. Without knowing the scale of the data, reporting only L1 loss makes evaluation difficult. Could authors additionally report the relative L2 loss?
12. As the authors acknowledge, the inference time of diffusion models limits their application, which is confirmed by the statistical results. The authors should consider using inference acceleration methods such as DDIM and verify that the method remains effective under DDIM sampling.

**Questions:**

1. The reviewer has some doubts about the selected generative model. For such prediction and simulation tasks, diffusion models are not always effective due to their SDE features. Flow matching, which is based on ODEs, appears more suitable for PDE solvers. Could the authors discuss the application of both models for PDE solvers and explain why they chose diffusion models over flow matching?
2. In line 229, "we find that GANs and perceptual guidance can in certain cases improve reconstruction performance, but for simplicity, we omit them in our main results." The author said GANs can improve the performance, but why you do not apply GANs?
3. In Figure 5, the middle two figures do not show turbulent flow, making this characterization inaccurate. Specifically, the second figure clearly shows laminar wake flow, and while the third figure shows vortices, it has not yet reached turbulent conditions.

---

> ### Author Response · Authors · 2024-11-25
> **Response to Reviewer zsUD**
>
> Thank you for the comprehensive review. Many of your comments have helped us to improve the quality of our work and which we greatly appreciate. Please find our response below (apologies in advance for how long it is!) and feel free to reach out if there are any additional concerns/questions.
>
> > The reviewer raises doubts about the novelty and contribution of the work, seemingly only using the encoding of the language model as a condition to generate solutions of PDEs.
>
> Thank you for raising this concern. In developing a text-conditioned neural solver, we needed to advance many aspects of diffusion models for PDEs. In particular this paper, to the best of our knowledge, is the first to do:
>
> - Full 4D spatiotemporal generation
> - Conditional latent diffusion for PDEs
> - Diffusion on arbitrary meshes for PDEs
>
> These new methods, especially for latent modeling and for solving PDEs on arbitrary meshes, extend the capabilities of diffusion models to more practical physics problems and improves their accuracy and speed with respect to previous autoregressive or pixel-space diffusion frameworks, which can be used beyond text2PDE applications. Within text-conditioned PDE generation, which this paper is the first to propose this problem, we release three new captioned datasets and show a framework for accomplishing a task that has yet to be done. More broadly, the work seeks to tackle a new problem and extend the capabilities of neural PDE solvers, which deviates from the norm of proposing new methods to advance a set of well-established benchmarks (which is still very useful!).
>
> > Why do the two experiments adopt different baselines and lack baselines such as FNO for cylinder flow?
>
> FNO is a regular grid method since it relies on regular spacing to evaluate the Fast Fourier Transform, as such it cannot be directly applied to mesh problems such as cylinder flow. In fact, the Unet, DilResNet, and ACDM benchmarks were all proposed for regular grid problems, and as a result, are not applicable to the cylinder flow problem. To remedy this, many studies have proposed adapting FNO to mesh problems, such as Geo-FNO [1] or GINO [2], and GINO is regarded as the more efficient and modern approach. As a result, we include GINO in our cylinder flow baseline, which can be seen as a proxy for FNO (indeed GINO uses FNO as a backbone).
>
> More broadly, including a diverse set of baselines ensures a fair and comprehensive comparison to current models. Our philosophy was to include benchmarks to compare against graph, attention, operator, convolution, and diffusion-based models to show the performance of the proposed LDM model, and adopting different baselines is both needed for the different problem domains as well as beneficial for a more holistic comparison.
>
> > From line 047 to 050, while this statement may seem obvious, it's not a scientific issue. It is necessary to have physics and numerical knowledge to build PDE solvers. At the same time, approaches based on FNO, DeepOnet and similar numerically-driven methods can learn accurate PDE solvers without requiring physics knowledge.
>
> Thank you for bringing this point up. While it is true that FNO/DeepOnet can be used without requiring physics knowledge, we believe a limiting factor to their adoption is in the opposite direction: in practical problems, engineers are limited by the deep learning expertise to properly train, evaluate, and query SOTA models. As technical problems are being solved (more accurate prediction, more generalizable, etc.) the main limitation is increasingly becoming the steep knowledge gap and inaccessibility of novel methods. This can also be seen in the market: new startups leveraging SOTA research generally serve as consultants or service providers to large engineering firms to deliver neural solver solutions due to the unique, interdisciplinary background needed to deploy these models. While this isn’t a purely scientific problem, it is still highly relevant for the practical use of neural solvers.
>
> To frame this in another light, image generation has been possible for many years and at very high quality. However, despite the technology and models being there, image generation was still largely only used within the CV community until text2image models were introduced; it is likely that a main factor for this is image generation models were highly inaccessible to practitioners before the use of a text modality. Artists, advertisers, or even just the general public are now able to leverage deep learning partly due to research targeted at the accessibility of image generation. We recognize that the introduction could use additional work to make this motivation more clear, and we appreciate your help in this regard.
>
> 1. Z. Li, et al., Fourier Neural Operator with Learned Deformations for PDEs on General Geometries
> 2. Z. Li, et al., Geometry-Informed Neural Operator for Large-Scale 3D PDEs. 2023. https://doi.org/10.48550/arXiv.2309.00583

---

> > ### Author Response · Authors · 2024-11-25
> > **Continued Response to Reviewer zsUD**
> >
> > > From line 053, this work deviates from the challenges described above, and there are concerns about potential overclaiming. Based on the author's two experiments, the proposed model only works in specific PDEs, spatiotemporal dimensions, and simple scenarios where the required physics knowledge is quite basic. In other words, the authors failed to address a crucial question: for complex PDE systems, how to represent the system using language, when mathematical formulas are clearly more suitable, while it is the main challenge the reviewer understand from the previous paragraph.
> >
> > Thank you for raising this point. The subsequent paragraph after line 053 is aimed at introducing text as a more interpretable and accessible modality for interacting with neural PDE solvers, which we can make more clear by rewriting the introduction (similar to the previous response). As for the next paragraph (starting from line 078), representing physics through language is an open question, and one that we don't want to claim to have solved. In fact, this paper is the first time this question has even been introduced.
> >
> > > Spatial and temporal resolution is crucial for PDE solvers. How does this work handle different spatial resolutions? Specifically, how does this method distinguish between initial conditions with different resolutions within the same system?
> >
> > Thank you for bringing this concern up. One of the main benefits of the proposed method is that it is able to handle inputs of arbitrary spatial discretization. In fact, the cylinder flow example has elements that are on the order of .001 meters up to .1 meters long, spanning two orders of magnitude in spatial discretization within a single frame of a solution and the proposed learnable aggregation/learnable interpolation is robust with respect to this input discretization and can handle varying discretizations across data samples. Within the cylinder flow benchmark, data samples can change from 1700 mesh points up to 2100 mesh points based on the geometry of the flow, and the proposed method can be universally applied without needing to construct a graph or use a separate data structure for each data sample. This performance in practice is also theoretically motivated. Using a latent grid with a learnable interpolation can be seen as a neural field [3], and regardless of the discretization, the method learns an underlying output field that can be queried arbitrarily and continuously.
> >
> > > Could you provide more details about the autoencoder in Sec. 3.1? This part is quite abstract and difficult for readers to understand intuitively. Please briefly explain the autoencoder's architecture, input/output structure, and training methodology. Additionally, during the diffusion model training, is the encoder trained simultaneously or are its parameters fixed?
> >
> > Happy to do this, we have provided more details in Section 3.1 as well as included more visualizations and explanations in Appendix B.3. During diffusion training, the autoencoder is frozen as described in line 257.
> >
> > > In line 235, the decoder appears to be CNN-based, which imposes constraints on the spatial dimensions of the output solutions. Furthermore, interpolation operations can lead to inaccuracies in the results, even when solutions on regular grids are accurate.
> >
> > It is correct that the decoder backbone is CNN-based, however the last few layers of the decoder are a learnable interpolation kernel to map from this regular grid to a mesh, which allows an arbitrary discretization of the outputs. This can be a highly accurate strategy, as shown in [3], and we also reproduce this result in the low reconstruction and prediction errors of the mesh autoencoder and LDM model. This is because the interpolated values are calculated from a learnable kernel integral, which can approximate arbitrary operators in its limit (and can be seen as a graph kernel neural operator) [4]. This learnable kernel integral can be highly expressive, nonlinear, and deep, and as a result, learns an accurate mapping from the grid to a continuous output function that can be evaluated at arbitrary query points as the output. Therefore, there are no constraints on the spatial dimensions of the output, and we clarify this in Appendix B.3.
> >
> > > In the method section, the authors have placed too many crucial details in the appendix, like Appendix C.2, B.1, B.2, D, making it impossible to understand the method without referring to the appendix.
> >
> > Apologies for this, and we appreciate your help in aiding us to improve the clarity of this work. We have shortened some of the background and added more details into the methods section. Unfortunately the page limits are quite strict and we opted to put more of the mathematical descriptions in the main body.
> >
> > 3. S. Wang, et al., CViT: Continuous Vision Transformer for Operator Learning
> > 4. Z. Li et al., Neural Operator: Graph Kernel Network for Partial Differential Equations

---

> > > ### Author Response · Authors · 2024-11-25
> > > **Continued Response to Reviewer zsUD**
> > >
> > > > The selected parameters for cylinder flow are overly simplistic, only covering laminar flow cases. Generally, the Kármán vortex street phenomenon is more common in cylinder flows, yet the authors did not include this scenario.
> > >
> > > The cylinder flow dataset does indeed contain Karman vortex streets, and we validate this by identifying samples within the dataset that have vortex shedding by looking at the temporal derivatives of data samples at later timesteps. If du/dt (change in velocity w.r.t. time) increases/remains large over time, we can conclude that periodic vortex shedding continues to occur, and if du/dt goes to zero over time, then the flow has reached a steady state. Interestingly, nearly half of the samples in the dataset contain Karman vortex streets, which is consistent with literature describing this phenomena occurring in Reynolds numbers from 300-2e5 [5], which is contained in the dataset’s Reynolds numbers of 60-1500. We include a discussion about this in Appendix E.1, and show a plot of the dataset’s Reynolds numbers and indicate samples where Karman vortex streets are identified.
> > >
> > > >All data used is 2D, and it's unclear whether this model can handle 1D and 3D experiments. The lack of such experiments significantly reduces the persuasiveness of the work.
> > >
> > > Thanks for raising this concern, we have included a new set of 3D experiments to test the proposed method on higher-dimensional, more complex PDE problems. In particular, we test on a 3D turbulence dataset and present details and results in Section 4.3. As for 1D experiments, these tend to be more toy problems and both current numerical and neural solvers are quite advanced to quickly and accurately solve these. As such, we seek to benchmark on more challenging and practically relevant settings.
> > >
> > > >Without knowing the scale of the data, reporting only L1 loss makes evaluation difficult. Could authors additionally report the relative L2 loss?
> > >
> > > Thank you for this suggestion, we have calculated the relative L2 Losses and changed L1 loss to relative L2 loss in the tables in the main text of the paper. (Tables 1 and 2)
> > >
> > > >As the authors acknowledge, the inference time of diffusion models limits their application, which is confirmed by the statistical results. The authors should consider using inference acceleration methods such as DDIM and verify that the method remains effective under DDIM sampling.
> > >
> > > Thank you for this suggestion. We have implemented DDIM sampling and presented results for this accelerated sampling method in Appendix D.3. DDIM sampling works quite well for PDE problems and is able to maintain high accuracies of the LDM while speeding up inference to a point where the LDM model can be faster than conventional, deterministic neural solvers. We provide a rationale for this as well, but briefly, we believe that this is the case since PDE solutions tend to have uni-modal conditional distributions, and as such, Langevin sampling converges very quickly which fewer sampling steps can take advantage of.
> > >
> > > 5. S. Chen, Flow-Induced Vibration of Circular Cylindrical Structures. 1985. doi:10.2172/6331788.

---

> > > > ### Author Response · Authors · 2024-11-25
> > > > **Continued Response to Reviewer zsUD**
> > > >
> > > > > The reviewer has some doubts about the selected generative model. For such prediction and simulation tasks, diffusion models are not always effective due to their SDE features. Flow matching, which is based on ODEs, appears more suitable for PDE solvers. Could the authors discuss the application of both models for PDE solvers and explain why they chose diffusion models over flow matching?
> > > >
> > > > We thank the reviewer for the insightful comment and helpful suggestion.
> > > > We agree that different diffusion formulations (i.e. different choices of $\alpha_t, \sigma_t$ given the forward process $q(\mathbf{x_t}|\mathbf{x})=\mathcal{N}(\mathbf{x_t}|\alpha_t \mathbf{x},\sigma_t^2 I)$), different prediction target and different sampler have varying strengths depending on the applications. More specifically, in this work we have studied the variance preserving process: $\alpha_t^2=1-\sigma_t^2$ and parameterize the model with epsilon prediction.
> > > >
> > > > We acknowledge that a detailed exploration of different diffusion setup for PDE prediction is indeed an interesting and valuable direction. However, for this study, we focused on showcasing the capabilities of the diffusion model under a unified text-conditioning framework for PDE prediction. As shown in prior work [6], different training targets in different diffusion models (including flow matching) can be converted into a unified form with different time-dependent weights. Therefore we believe other diffusion paths can also achieve reasonable accuracy using our framework.
> > > >
> > > > To demonstrate that our proposed method can also work with deterministic ODEs, we provide the results of our model using a DDIM sampler (which is shown to be the first-order discretization of the probability flow ODE under change-of-variables [7]), and show the results in Appendix D.3. The results show that the proposed framework is amenable to be used with different samplers and can achieve similar accuracies regardless of the diffusion path.
> > > >
> > > > > In line 229, "we find that GANs and perceptual guidance can in certain cases improve reconstruction performance, but for simplicity, we omit them in our main results." The author said GANs can improve the performance, but why you do not apply GANs?
> > > >
> > > > Thank you for raising this question. You are correct in pointing out that GANs can improve reconstruction performance, but we found this not to always be true. In particular, for the smoke buoyancy dataset, GAN guidance seemed to destabilize training and lead to larger validation errors when training the autoencoder. This is because the smoke buoyancy dataset is more complex and contains finer features; as a result, the discriminator was often unable to distinguish real from fake samples. With more tuning and more complex GAN techniques, this could eventually work but to maintain a consistent and more simple model pipeline across both experiments, we opt to omit adversarial training from the main results.
> > > >
> > > > > In Figure 5, the middle two figures do not show turbulent flow, making this characterization inaccurate. Specifically, the second figure clearly shows laminar wake flow, and while the third figure shows vortices, it has not yet reached turbulent conditions.
> > > >
> > > > Thank you for pointing this out, we also agree with this assessment and will provide a discussion about this. This issue stems from the fact that we use simple cutoffs based on Reynolds numbers to generate this description in the prompt. While the flow regimes around a cylinder have been well-studied, there are still varying standards to characterize these flow regimes based on Reynolds numbers. For example, from Blevins, 1990, the flow regimes for Karman vortex streets are presented as: 40-150 (laminar), 150-300 (transition), 300-3e5 (turbulent) [8]. Many works have also seeked to refine these regimes, up to even 15 different flow regimes and using CFD to also characterize the problem. The upshot is that these regimes can be very problem dependent; in our case, the location of the cylinder affects the resulting flow (cylinders close to the boundary remain laminar at higher Reynolds numbers), and simple cutoffs based on Reynolds numbers may not be accurate for all cases. While we recognize this is a limitation due to not being able to manually label data, the precise semantics of this portion of the text prompt do not seem to largely affect the results; however, refining the qualitative description of the flow is a promising direction to further improve the model.
> > > >
> > > > 6. Kingma, D., & Gao, R. (2024). Understanding diffusion objectives as the elbo with simple data augmentation. Advances in Neural Information Processing Systems, 36.
> > > > 7. Lu, Cheng, et al. "Dpm-solver: A fast ode solver for diffusion probabilistic model sampling in around 10 steps." Advances in Neural Information Processing Systems 35 (2022): 5775-5787.
> > > > 8. R.D. Blevins,  Flow-induced vibration. 1990.

---

> > > > > ### Comment · Reviewer_zsUD · 2024-11-26
> > > > >
> > > > > Thanks for your response. The new experiments on 3D setting and DDIM have addressed my questions about them. Other parts of my concerns, including baselines, resolutions, interpolation, and presentation, have been addressed by the response. However, the concerns in other parts still exist, among which the two most important ones are:
> > > > >
> > > > > 1. There is no evidence to suggest that this new proposed problem in the work is significant, as there is no reason to describe it in text for a PDE system that can be represented by strict mathematics equations. Although the author emphasizes its usage in special scenarios, the current version is too far from the scenario declared by the author. Engineering firms often require models with stronger generalizations, such as how to generalize multiple cylinders, cubes, and cylinder rotations. Therefore, a foundation model is feasible to solve the author's proposed problem, but there is still a significant gap between the current work and the foundation model. I consider that even if this problem needs to be solved, this work has not yet been accomplished.
> > > > >
> > > > > 2. Regarding another concern of novelty, conditional latent generation has been applied in many fields including image, video, and text. The reviewer does not consider that applying methods from other fields to the PDE field is an innovative approach.
> > > > >
> > > > > Based on the above discussion, I would like to maintain my initial rating.

---

> ### Author Response · Authors · 2024-12-02
>
> Thank you for the reply, and we are happy to have addressed a large portion of your concerns. The proposed problem is highly novel and perhaps the significance of it has yet to be fully explored or understood, and we respect your opinion. As for the execution, it was already quite challenging to produce results at this scale (needing to make new datasets, new mesh auto-encoder methods, new evaluation metrics, large compute/engineering effort, etc.) and creating a foundation model could be an avenue for future work since this is the first time a problem like this has been proposed or attempted to be solved.
>
> As for the second point, there are many works (including those published at ICLR) that could be considered as applying methods from other fields to PDEs [1,2,3] (a non-exhaustive set of examples that have been influential). While the acceptable degree of this crossover is subjective and can be influenced by other factors (execution, presentation, standards in the field, etc.), at a certain point all scientific works build on each other, and indeed we seek to extend, rather than replicate, previous methods to PDEs by introducing novel mesh-based diffusion and higher dimensionality (4D) sample generation, as well as introducing a new direction of research. However, we respect your views and leave this interpretation up to the community.
>
> 1. Tobias Pfaff, et al., Learning Mesh-Based Simulation with Graph Networks. ICLR, 2021, https://arxiv.org/abs/2010.03409 (applying GNNs to physics simulation)
> 2. Tung Nguyen et al., ClimaX: A foundation model for weather and climate. ICML, 2023, https://arxiv.org/abs/2301.10343 (applying ViT + scaling to climate prediction)
> 3. Kimberly Stachenfeld., et al., Learned Coarse Models for Efficient Turbulence Simulation. ICLR, 2022, https://arxiv.org/abs/2112.15275 (applying CNN/Dil-Resnet to Turbulence simulation)

---

### Official Review · Reviewer_Somp · 2024-10-30

**Soundness:** 3
**Presentation:** 3
**Contribution:** 2
**Rating:** 6
**Confidence:** 3

**Summary:**

This paper proposes a surrogate neural network model that solves PDE problems. The authors claim innovations in three parts: mesh en/decoder, latent physics solver, and conditioning. The mesh en/decoder unifies differently discretized solution data achieved from various numerical solvers. Unlike an autoregressive model, the latent physics solver utilizes a diffusion model and predicts a full spatiotemporal solution. For the conditioning, two mechanisms, i.e., state-based and text-based methods, are proposed.

**Strengths:**

Under the aim of developing a surrogate neural PDE solver, the paper claims three major novelties (also with their strengths) of the proposed model.

1. En/decoding PDE data via a uniform spatiotemporal latent grid: Regardless of discretization, the PDE data is first encoded into the uniform grid via a weighted sum using a learned parameterized kernel. Its architecture unification is beneficial, and thanks to the benefit, the conventional CNN can outperform GNN and Neural Fields in training an autoencoder for irregular data.

2. Latent diffusion model for entire solution trajectory prediction: Instead of a state prediction of the autoregressive model, the proposed neural PDE solver aims to predict the trajectory given initial and boundary conditions. This generative model can mitigate error accumulation, and the work compares with other models.

3. Text conditioning: In addition to the traditional physics-state initial and boundary conditions, the paper proposes to use natural language as a means of conditioning. The authors claim that this can be a promising direction for PDE simulators.

Overall, with the combination of these novelties, the paper claims their surrogate PDE solver outperforms the other models such as GINO, OFormer, FNO, ACDM, etc.

Additionally, thanks to the accompanying codes, the results should be reproducible (although I did not run the codes myself).

---

***During the discussion,*** the authors clarified my concerns; thus I raised my score to 6 from 5.

**Weaknesses:**

As the title "text2pde" stands for, the paper proposes a text-based conditioning for PDE. Although it is an interesting direction to investigate, the practical benefit of the current model is not fully convincing to me. If I am not wrong, the text-based conditioning of the proposed method eventually relies on a set of extracted values that are necessary to set up the initial condition, and this is only for conditioning a single initial frame. In my view, this can be easily and more precisely defined with a script-based (or template-based) initialization. Arguably, the text-based approach may rather be an ambiguous means to use in this context. Moreover, this part looks independent of the solver; it tackles how to generate an image (i.e., a PDE state in this context) from a text, which looks closer to text2image rather than text2pde.

The validation of the proposed solver could be further improved. Only two examples of N.-S. equations were investigated. The proposed entire-trajectory prediction was only evaluated for 25 steps in the cylinder flow case with low turbulence, which is relatively less complex, and 48 steps in the buoyancy-driven flow. Further experiments for the multi-step trajectory prediction would improve the validation.

**Questions:**

The example of Cylinder Flow describes that the physical variables are velocity and pressure. Do the reported L1 Loss values take into account both variables? If so, the errors of both variables should be discussed more thoroughly.

Similarly, it would be worth having further clarification of the error for two variables (velocity and density) in the Buoyancy-driven Flow example.

In Buoyancy-driven Flow, the samples vary with not only initial conditions but also buoyancy factors. Does the model get conditioned with the buoyancy factor? I guess not. Then, the trajectories (evolved with different buoyancy factors) represent somewhat different physics. In this case, I am not sure how the model has learned the underlying physics.

It would be valuable to see how the radius of a spatio-temporal ball affects the model performance.

It would be helpful to clarify how to define/train the kernel $\kappa(\cdot)$.

---

> ### Author Response · Authors · 2024-11-25
> **Response to Reviewer Somp**
>
> Thank you for the comprehensive review. Please find our response below and feel free to reach out if there are any additional concerns.
>
> > As the title "text2pde" stands for, the paper proposes a text-based conditioning for PDE. Although it is an interesting direction to investigate, the practical benefit of the current model is not fully convincing to me. If I am not wrong, the text-based conditioning of the proposed method eventually relies on a set of extracted values that are necessary to set up the initial condition, and this is only for conditioning a single initial frame. In my view, this can be easily and more precisely defined with a script-based (or template-based) initialization. Arguably, the text-based approach may rather be an ambiguous means to use in this context. Moreover, this part looks independent of the solver; it tackles how to generate an image (i.e., a PDE state in this context) from a text, which looks closer to text2image rather than text2pde.
>
> Thank you for raising this point and for providing this perspective. There are a couple of concerns bundled together that we can address.
>
> Firstly, to motivate the practical merit/contribution from a technical standpoint, in order for text-conditioned generation to work, we needed to advance many aspects of diffusion models for PDEs. Specifically, this is the first work to do:
>
> - Full 4D spatiotemporal generation
> - Conditional latent diffusion
> - Diffusion on arbitrary meshes
>
> for PDE problems, which could be of practical benefit even beyond text2PDE applications.
>
> Secondly, to motivate the need for natural language rather than simply using extracted values, we perform a set of ablation studies (Table 7, Appendix D.1) that investigate conditioning on a vector of the extracted values—cylinder radius, cylinder position, and Reynolds number—(Vector-Ablate) as well as conditioning on these values as a text prompt without the surrounding context (Prompt-Ablate). We find that using the full, semantically meaningful prompt has 2-4 times lower errors than these ablations and has nearly the same the error of using the initial field of physics values. While in theory a script/template based initialization using these extracted values can describe the flow, no neural or even numerical solver does this in practice and works quite poorly due to the sparsity of the initialization. The practical contribution of natural language can be seen as a more compact, meaningful, and interpretable representation than the initial flow field, as the text prompt has only 50 words (1Kb of data) compared to about 2000 velocity and pressure floats (1Mb of data).
>
> More broadly, we believe that text-conditioned generation can have practical benefits for the real-world adoption of neural solvers. We believe that neural PDE solvers are becoming increasingly limited by a knowledge/accessibility gap when it comes to their practical application. As technical problems are being solved (more accurate prediction, more generalizable, etc.) the main limitation is increasingly becoming the steep knowledge gap and inaccessibility of novel methods. This can also be seen in the market: new startups leveraging SOTA research generally serve as consultants or service providers to large engineering firms to deliver neural solver solutions due to the unique, interdisciplinary background needed to deploy these models. While this isn’t a purely scientific problem, it is still highly relevant for the practical use of neural solvers, and creating more interpretable, accessible neural solvers can help practitioners use better novel tools to solve challenges.
>
> To frame this in another light, image generation has been possible for many years and at very high quality. However, despite the technology and models being there, image generation was still largely only used within the CV community until text2image models were introduced; it is likely that a main factor for this is image generation models were highly inaccessible to practitioners before the use of a text modality as a query. Artists, advertisers, or even just the general public are now able to leverage deep learning partly due to research targeted at the accessibility of image generation.
>
> Lastly, for the text2image distinction, we recognize that many of the prompts generally describe an initial state for the simulation. However, in practice, this is the only information we have about PDE systems when querying a model; indeed when setting up a numerical simulation we can only specify an initial set of parameters to evolve a solution. However, these initial parameters still greatly influence the resulting solution, and therefore coupling the prompt to the resulting spatio-temporal generation ensures consistency between the user query and the solver solution.

---

> > ### Author Response · Authors · 2024-11-25
> > **Continued Response to Reviewer Somp**
> >
> > > The validation of the proposed solver could be further improved. Only two examples of N.-S. equations were investigated. The proposed entire-trajectory prediction was only evaluated for 25 steps in the cylinder flow case with low turbulence, which is relatively less complex, and 48 steps in the buoyancy-driven flow. Further experiments for the multi-step trajectory prediction would improve the validation.
> >
> > Thank you for raising this concern. We have run an additional experiment to investigate multi-step trajectory prediction, up to 100 timesteps for the cylinder flow problem by adapting the proposed LDM method to be queried auto-regressively, shown in Appendix C.4. This also means that the proposed approach can in theory generalize to arbitrary time horizons. Furthermore, we improve the validation by including a new experiment on a 3D Turbulence dataset to evaluate the performance and scalability of the model in complex, large-scale PDE problems, in Section 4.3.
> >
> > > The example of Cylinder Flow describes that the physical variables are velocity and pressure. Do the reported L1 Loss values take into account both variables? If so, the errors of both variables should be discussed more thoroughly.
> >
> > Thank you for this question. The reported losses do indeed take into account both variables, and we will discuss the errors in the predicted velocity and pressure in the Results. In general, we find that the velocity and pressure errors are about equal in magnitude, with pressure having slightly higher errors. This trend is consistent across all LDM model variants. We make a note about this in the results of Section 4.1.
> >
> > > Similarly, it would be worth having further clarification of the error for two variables (velocity and density) in the Buoyancy-driven Flow example.
> >
> > Thank you for this point. We have included a clarification between the differences between the velocity and density errors in the buoyancy-driven flow problem in the Results section. We find that the errors for velocity are larger than the errors for density, around 1.5 times higher, which is consistent across all LDM model variants as well. We make a note about this in the results of section 4.2.
> >
> > > In Buoyancy-driven Flow, the samples vary with not only initial conditions but also buoyancy factors. Does the model get conditioned with the buoyancy factor? I guess not. Then, the trajectories (evolved with different buoyancy factors) represent somewhat different physics. In this case, I am not sure how the model has learned the underlying physics.
> >
> > Thank you for raising this point. In the buoyancy-driven flow problem, the samples indeed are conditioned on the buoyancy factors. For the text-conditioned generation, the buoyancy factor is appended to the prompt with the string “the buoyancy factor is {}”, and for the IC-based generation, the buoyancy factor is given to the DiT backbone as an additional vector using Adaptive Instance Normalization for global conditioning. Similarly, the baselines also receive the buoyancy factor information through adaptive instance normalization. We apologize for the confusion.
> >
> > > It would be valuable to see how the radius of a spatio-temporal ball affects the model performance.
> >
> > We have run these experiments and put the results in Appendix B.1, Table 4a. In general, increasing the ball radius slightly reduces autoencoder reconstruction loss.
> >
> > >It would be helpful to clarify how to define/train the kernel.
> >
> > We have included additional details on the kernel aggregation/interpolation in Appendix B.3, including two new figures to explain this. Thank you!

---

> > > ### Comment · Area_Chair_xkQ5 · 2024-11-26
> > >
> > > Dear reviewer,
> > >
> > > Please make to sure to read, at least acknowledge, and possibly further discuss the authors' responses to your comments. Update or maintain your score as you see fit.
> > >
> > > The AC.

---

> > > > ### Comment · Reviewer_Somp · 2024-11-26
> > > >
> > > > I appreciate the authors making a great effort to make a revision and answer my questions. Many of my concerns have been addressed, particularly, it clarified my initial misunderstanding about "text conditioning". Thus, I raised my score to 6 from 5.
> > > >
> > > > Nonetheless, some concerns still remain. As the authors also claim the value of other contributions in improving the diffusion model for PDE, the current title (in particular, Text2PDE) is somewhat misleading in my opinion. Additionally, as the experiments demonstrate, the first frame conditioning (LDM-FF) seems to always outperform the text conditioning (LDM-Text). Then, why doesn't the model use a text2image model that can generate the first frame from text and then use it for LDM-FF?

---

> > > > > ### Author Response · Authors · 2024-12-02
> > > > >
> > > > > Thank you for the reply and for the constructive feedback! Unfortunately it seems as if we have run out of time to run the text2img + solver baseline, however it is a great suggestion. Despite not having concrete results, there is some intuition from prior works comparing text2img + temporal rollout/inflation vs. full text2video generation that seem to suggest that the full generation is more temporally consistent [1]. To mitigate this, some auto-regressive text2video generators use an "anchor/reference frame" to ensure the consistency of the auto-regressive generator with respect to the initial condition [2]. Regardless, we are happy to have addressed your other concerns and we appreciate that the score has reflected this.
> > > > >
> > > > > 1. Omer Bar-Tal, et al., Lumiere: A Space-Time Diffusion Model for Video Generation, https://arxiv.org/abs/2401.12945
> > > > > 2. Wenming Weng, et al., ART⋅V: Auto-Regressive Text-to-Video Generation with Diffusion Models, https://arxiv.org/abs/2311.18834

---

### Official Review · Reviewer_jPnk · 2024-11-01

**Soundness:** 3
**Presentation:** 3
**Contribution:** 3
**Rating:** 6
**Confidence:** 5

**Summary:**

Very new paper. I support this because it actually works, I appreciate the authors working! Just like language model for catalyst. But I just do not see why people need this as they need for catalyst-gpt. The authors know fluid, they do not need this even they can make this fluid-gpt. For people do not know fluid, why they need it?

**Strengths:**

It is first time people to get-like fluid sequence generation, to combine llm with diffusion is good.

**Weaknesses:**

The motivation is lacking. The flow simulation is not physical.

**Questions:**

The author should write down more about when, who will need this method to do what. And the flow past cylinder data seems under-resolved. The mesh is too coarse, can we make it finer?

---

> ### Author Response · Authors · 2024-11-25
> **Response to Reviewer jPnk**
>
> Thank you for the review and we are glad that you support this work. Please find our response below and feel free to reach out if there are any further questions/comments.
>
> > The motivation is lacking. The flow simulation is not physical.
>
> Thank you for raising this concern. We have expanded on the motivation in the introduction, but to provide a brief summary, we believe that neural PDE solvers are becoming increasingly limited by a knowledge/accessibility gap when it comes to their practical application. As technical problems are being solved (more accurate prediction, more generalizable, etc.) the main limitation is increasingly becoming the steep knowledge gap and inaccessibility of novel methods. This can also be seen in the market: new startups leveraging SOTA research generally serve as consultants or service providers to large engineering firms to deliver neural solver solutions due to the unique, interdisciplinary background needed to deploy these models. While this isn’t a purely scientific problem, it is still highly relevant for the practical use of neural solvers.
>
> To frame this in another light, image generation has been possible for many years and at very high quality. However, despite the technology and models being there, image generation was still largely only used within the CV community until text2image models were introduced; it is likely that a main factor for this is image generation models were highly inaccessible to practitioners before the use of a text modality. Artists, advertisers, or even just the general public are now able to leverage deep learning partly due to research targeted at the accessibility of image generation.
>
> As for the second point, neural and even numerical solvers often have trouble maintaining physical consistency in generated solutions, but the proposed work is able to have accuracies that outperform neural baselines at a faster speed than numerical baselines. We believe that the physical consistency can be improved by doing a full spatio-temporal prediction, which reduces the amount of calls to the solver and delays error accumulation.
>
> > The author should write down more about when, who will need this method to do what. And the flow past cylinder data seems under-resolved. The mesh is too coarse, can we make it finer?
>
> Thank you for raising this question. We have improved the introduction to better motivate and situate the contribution and the method. As for the second point, the cylinder-flow dataset is from Pfaff et al. (https://arxiv.org/abs/2010.03409), and unfortunately we do not have access to the COMSOL simulations used to generate this data in order to refine it. However, this dataset is quite common and has been used in many subsequent works in the PDE field.

---

> > ### Comment · Area_Chair_xkQ5 · 2024-11-26
> >
> > Dear reviewer,
> >
> > Please make to sure to read, at least acknowledge, and possibly further discuss the authors' responses to your comments. Update or maintain your score as you see fit.
> >
> > The AC.

---

> > > ### Author Response · Authors · 2024-11-29
> > >
> > > Dear reviewer,
> > >
> > > Just a gentle reminder that the last day to make a comment is December 2nd. If there are any additional questions or concerns we would be happy to address them. Thank you !

---

> > ### Comment · Reviewer_jPnk · 2024-12-02
> > **I support this paper.**
> >
> > It is new, no one did this before.

---

> > > ### Author Response · Authors · 2024-12-02
> > >
> > > Thank you for your support and for noting the work's novelty. We hope that we were able to address your concerns!

---

### Official Review · Reviewer_7eso · 2024-11-03

**Soundness:** 3
**Presentation:** 3
**Contribution:** 2
**Rating:** 6
**Confidence:** 5

**Summary:**

The authors introduce a new approach to physics simulations using latent diffusion models. Their method is anchored by (i) the use of mesh autoencoders, (ii) full spatio-temporal modeling to avoid autoregressive error accumulation, and (iii) conditioning on text prompts.

**Strengths:**

- Latent space modeling for physics simulations. By using latent diffusion models, the authors effectively compress and encode high-dimensional physics data into a lower-dimensional latent space. Such latent space approaches contribute to more adaptable and faster simulations, making it a powerful alternative to traditional numerical solvers. There is a surge of latent space PDE models, this paper fits well into this paradigm. The full spatio-temporal diffusion modeling conditioned on text prompts and the subsequent decoding stands out as unique modeling approach.

- Text prompts for simulation control. Text2pde is an interesting direction since it allows users to define simulation parameters or initial conditions using simple, interpretable text commands. The potential for language-based input makes physics simulations more accessible and generalizable. Yet it is not clear what is the difference to simple parameter modulation, the nice capturing of the initial condition is cool.

- Application to mesh-based dynamics.

**Weaknesses:**

- Text2pde seems to be hard to scale to larger problems. To avoid confusion: scalability here refers to input representations. The mesh autoencoder is a good direction, yet the latent space is a latent grid, which if going to 3D is fundamentally limiting. Also for larger 2D grids, a tokenized representation might be preferable. Secondly, the generation of the full trajectory is storage intensive, and for 3D problems would result in a 4D tensor, which is computationally infeasible. Most turbulent problems, where neural surrogates might shine are 3D, presenting a drawback of the applicability of the method.
- Text conditioning in this framework seems to be a slightly sophisticated parameter conditioning / parameter modulation. The nice feature so far is the generation of the initial mesh, initial condition. It would greatly help to make these differences clearer. Further the text prompt part needs to be adapted for each problem individually. It should be clarified how the text part can maybe be used across settings? Is this even possible?

**Questions:**

- Autoregressive vs entire solutions. It is not clear whether producing an entire solution trajectory is really beneficial. First of all it doesn’t scale to 3D problems or larger 2D problems. Secondly, it might be unphysical. Do the authors have insights on this?

---

> ### Author Response · Authors · 2024-11-25
> **Response to Reviewer 7eso**
>
> Thank you for your thoughtful review, we have performed some additional experiments to help improve our work. Please find our response below and feel free to reach out if there are any additional questions/concerns.
>
> > Text2pde seems to be hard to scale to larger problems. To avoid confusion: scalability here refers to input representations. The mesh autoencoder is a good direction, yet the latent space is a latent grid, which if going to 3D is fundamentally limiting. Also for larger 2D grids, a tokenized representation might be preferable. Secondly, the generation of the full trajectory is storage intensive, and for 3D problems would result in a 4D tensor, which is computationally infeasible. Most turbulent problems, where neural surrogates might shine are 3D, presenting a drawback of the applicability of the method.
>
> Thank you for raising this concern, and we have performed an additional experiment to address this. In particular, we have investigated scaling this model to a 3D Turbulence problem where we spatio-temporally generate a 4D output. Our latent modeling approach, as well as using a DiT which internally tokenizes the latent grid, is able to make this computationally tractable, and we present results in Section 4.3 of the revised manuscript. Additionally, to scale the model to longer prediction timelines, we also present results where we modify the proposed LDM approach to work autoregressively and test this modified approach on a longer time-horizon in Appendix C.4).
>
> > Text conditioning in this framework seems to be a slightly sophisticated parameter conditioning / parameter modulation. The nice feature so far is the generation of the initial mesh, initial condition. It would greatly help to make these differences clearer. Further the text prompt part needs to be adapted for each problem individually. It should be clarified how the text part can maybe be used across settings? Is this even possible?
>
> Thank you for raising this concern. While text conditioning could be viewed as a parameter modulation, we believe that having semantically meaningful prompts is still helpful and provide some ablation studies to justify this (Table 7, Appendix D.1). In summary, we study conditioning on a vector of the normalized parameters—cylinder radius, cylinder position, and Reynolds number—(Vector-Ablate) as well as conditioning on these parameters as a text prompt without the surrounding context (Prompt-Ablate). We find that using the full, semantically meaningful prompt has 2-4 times lower errors than these ablations and has nearly the same the error of using the initial field of physics values. While in theory only a few parameters can describe the flow, no neural or even numerical solver does this in practice, and the natural language prompt in this case can even be seen as a more compact, meaningful representation than the initial flow field, as the text prompt has only ~50 words (~1Kb of data) compared to about 2000 velocity and pressure floats (~1Mb of data).
>
> As for the mesh, the decoder actually uses the provided mesh to collocate the latent solution, however the diffusion model does not. Therefore the diffusion backbone still generates coherent solutions (based on the prompt or first frame) even when the decoder is provided with an incorrect or arbitrary mesh, and illustrate this discretization invariance in Appendix B.3.
>
> Lastly, to use text-prompts across datasets/problems we and the field as a whole currently don’t have the datasets to accomplish this. Since this is the first time text2PDE is being investigated, generalizing text-conditioned solvers across problems would be an important future direction, however, for this current work we show that this concept is achievable and provide a working framework for future researchers to expand on. However, as long as a dataset can be captioned across different problems, which should be possible due to the expressivity of natural language, the current framework should be applicable.

---

> ### Author Response · Authors · 2024-11-25
> **Continued Response to Reviewer 7eso**
>
> > Autoregressive vs entire solutions. It is not clear whether producing an entire solution trajectory is really beneficial. First of all it doesn’t scale to 3D problems or larger 2D problems. Secondly, it might be unphysical. Do the authors have insights on this?
>
> Thank you for this question. We have performed an additional set of experiments to explore autoregressive generation by using the LDM framework autoregressively, albeit with a large context window. We describe this approach and provide results in Appendix C.4 of the revised manuscript.
>
> As for the second point, we believe that larger spatio-temporal predictions can help with maintaining physical consistency in the predicted solution, as even autoregressive neural solvers can produce unphysical results. In particular, spatio-temporal generation is able to delay error accumulation by having fewer model calls [1]. Furthermore, spatio-temporal generation is often used in the video diffusion domain to ensure temporal consistency between video frames [2], rather than using an image model autoregressively which would require separate approaches to semantically align individual frames to ensure consistency [3].
>
> 1. Johannes Brandstetter, Daniel Worrall, Max Welling, Message Passing Neural PDE Solvers, https://arxiv.org/abs/2202.03376
>
> 2. Omer Bar-Tal, Hila Chefer, Omer Tov, Charles Herrmann, Roni Paiss, Shiran Zada, Ariel Ephrat, Junhwa Hur, Guanghui Liu, Amit Raj, Yuanzhen Li, Michael Rubinstein, Tomer Michaeli, Oliver Wang, Deqing Sun, Tali Dekel, Inbar Mosseri, Lumiere: A Space-Time Diffusion Model for Video Generation, https://arxiv.org/abs/2401.12945
>
> 3. Andreas Blattmann, Robin Rombach, Huan Ling, Tim Dockhorn, Seung Wook Kim, Sanja Fidler, Karsten Kreis, Align your Latents: High-Resolution Video Synthesis with Latent Diffusion Models, https://arxiv.org/abs/2304.08818

---

> > ### Comment · Reviewer_7eso · 2024-11-26
> > **Thanks for the rebuttal**
> >
> > I do think this is an important contribution and have thus raised my score.

---

> > > ### Author Response · Authors · 2024-12-02
> > >
> > > Thank you for updating your score and we appreciate your thoughts on its importance!

---

### Official Review · Reviewer_9pTM · 2024-11-04

**Soundness:** 1
**Presentation:** 3
**Contribution:** 2
**Rating:** 5
**Confidence:** 3

**Summary:**

The paper introduces a framework called TEXT2PDE, which employs text-conditioned latent diffusion models (LDMs) to model spatio-temporal PDE trajectories. The authors introduce a mesh autoencoder that can encode and decode physical fields on arbitrary meshes. Spatio-temporal trajectories are generated using a conditional denoising process of an LDM with the initial conditions and boundary conditions of the PDE imposed into the conditioning sequence of the denoising backbone. A main novelty is the introduction of text conditioning, by employing a pre-trained transformer encoder. The authors argue that using natural language as a conditioning modality can improve the interpretability and usability of neural PDE solvers.

**Strengths:**

1. Thanks to the encoding of arbitrary meshes, the authors are able to apply their method to complex geometries – an important use case for the application of PDEs.
1. The main strength of the paper is the introduction of text conditioning for generation of PDE trajectories. The idea is novel and opens up a new research direction.
However, it would be still interesting to see more specific motivating examples of where text conditioning could be useful in real-world use cases of PDE solvers.

**Weaknesses:**

1. The evaluations seem lacking when assessing the text conditioning capability of the Text2PDE model. Firstly, the initial and boundary conditions of the cylinder dataset can be entirely encoded using just three real numbers, making the expressiveness of natural language excessive for this case. Secondly, the smoke dataset appears to be a poor fit for text-based description, as the smoke plumes are too unconstrained and unstructured to be accurately captured by language. (See also the question 1.)

2. In my opinion, evaluating text conditioning would require a new dataset with more regular shapes placed in general positions. The configurations should be rich enough to necessitate natural language for description, but constrained enough that a short text can accurately describe the image. Currently, it seems that the TEXT2PDE system is limited to using text conditioning to generate only the initial condition. This could be achieved more directly by combining text-to-image techniques and then running a standard (neural) PDE solver. The added benefit of using a combined system is not clear. It could be more interesting (and useful) if the PDE dynamics were also informed by natural language—perhaps through a text-encoded equation.

**Questions:**

1. As the text prompts appear to accurately capture the first frame of the reference (as shown in Table 2 and Figure 4 in Appendix A), I wonder if the model is memorizing and retrieving the smoke plume from the training dataset that most closely matches the input. For the existing prompts, would it be possible to display the most similar first frame from the training set?

---

> ### Author Response · Authors · 2024-11-25
> **Response to Reviewer 9pTM**
>
> Thank you for taking the time to compose the review and for the feedback. Please see our response below and feel free to reach out if there are any further questions/concerns.
>
> > The evaluations seem lacking when assessing the text conditioning capability of the Text2PDE model. Firstly, the initial and boundary conditions of the cylinder dataset can be entirely encoded using just three real numbers, making the expressiveness of natural language excessive for this case. Secondly, the smoke dataset appears to be a poor fit for text-based description, as the smoke plumes are too unconstrained and unstructured to be accurately captured by language. (See also the question 1.)
>
> Thank you for raising this concern. We would also agree that the cylinder dataset is highly constrained, however, we provide some ablation studies to justify the need for natural language (Table 7, Appendix D.1). In summary, we study conditioning on a vector of the normalized cylinder radius, cylinder position, and Reynolds number (Vector-Ablate) as well as conditioning on these values as a text prompt without the surrounding context (Prompt-Ablate). We find that using the full, semantically meaningful prompt has 2-4 times lower errors than these ablations and has nearly the same the error of using the initial field of physics values. While in theory only three numbers can describe the flow, no neural or even numerical solver does this in practice, and the natural language prompt in this case can even be seen as much less expressive than the initial flow field, as the text prompt has only 50 words (1Kb of data) compared to about 2000 velocity and pressure floats (1Mb of data).
>
> As for the second concern, we would also agree that the smoke dataset is not the best-suited for an accurate description by language. While this is true, we believe it doesn’t necessarily limit the usefulness of the model or the insight of the paper. One way to see this is that text2video or text2img models face a similar unconstrained and unstructured problem setup, yet models still are required to learn underlying semantic meaning. In our case, while the smoke plumes are quite unstructured (indeed the initial densities are sampled from a mixture of Gaussians), the model is still able to learn the underlying physics to evolve the generated solutions in a consistent manner. We believe that this still demonstrates the utility and promise of this method, and if more precise simulations are needed, the model can easily be used in the -FF mode for more precise results. While in the initial manuscript the LDM-FF and LDM-Text models were separate, we introduce new results where we train a single model that can switch between text and first-frame conditioning by padding the unused conditioning modality with zeros, in Appendix C.4.

---

> > ### Author Response · Authors · 2024-11-25
> > **Continued Response to Reviewer 9pTM**
> >
> > > In my opinion, evaluating text conditioning would require a new dataset with more regular shapes placed in general positions. The configurations should be rich enough to necessitate natural language for description, but constrained enough that a short text can accurately describe the image. Currently, it seems that the TEXT2PDE system is limited to using text conditioning to generate only the initial condition. This could be achieved more directly by combining text-to-image techniques and then running a standard (neural) PDE solver. The added benefit of using a combined system is not clear. It could be more interesting (and useful) if the PDE dynamics were also informed by natural language—perhaps through a text-encoded equation.
> >
> > Thank you for raising this concern, there are a couple of points to address within this comment. For the new dataset with regular shapes, we have run a new experiment to address this. In particular, we have run experiments and baselines on a 3D turbulence dataset with 45 various shapes (squares, rectangles, steps, bars, elbows, donut, U-shape, T-shape, H-shape etc.) with highly complex flow patterns, and also manually captioned this dataset which is more amenable to text descriptions. For a full picture of all the shapes, we recommend the last page of Lienen et al. (https://arxiv.org/abs/2306.01776)
> >
> > As for using PDE dynamics in the prompt, we considered this but were limited by the fact that the dynamics are rarely known a priori during inference. We can only make very broad time-dependent claims (i.e., the flow is turbulent, laminar, etc.) and the equation is the same in all samples (coefficients don’t change, using incompressible Navier-Stokes).
> >
> > Lastly, using text2img coupled with a PDE solver seems quite interesting and could be a direction for future work. However, one reason that we decided on the current method is the training difficulties that could arise when decoupling the solution process. In particular, the generative text2img process seeks to generate an initial condition that matches the description yet this can lead to multiple viable initial flow fields (for example in smoke buoyancy) or even small fluctuations in the sampled initial condition. Therefore, when training the neural solver, which has only seen perfect samples from the training set, the performance may suffer during inference, whereas when predicting the full solution at once it is easier to ensure consistency between the prompt, sampled initial condition, and generated solution.
> >
> > > As the text prompts appear to accurately capture the first frame of the reference (as shown in Table 2 and Figure 4 in Appendix A), I wonder if the model is memorizing and retrieving the smoke plume from the training dataset that most closely matches the input. For the existing prompts, would it be possible to display the most similar first frame from the training set?
> >
> > Thank you for the question, we have anonymously posted an image of the closest training samples to the validation set here, since the manuscript has gotten quite long and we are unsure of how to fit this in: https://i.postimg.cc/t4d0jcV7/train-closest.png
> >
> > The training set images are quite close to the generated samples, which in this case seems to be a limitation of the given dataset (from a separate paper) and the captioning process. The validation samples themselves are quite similar to the training set, and using an LLM to caption the two data splits could also result in similarities in the captions as well. However, we can notice that the solutions do diverge over time, and eventually reach noticeably different states. While the initial conditions are quite similar, small deviations eventually cause differences that the model would need to accurately model.

---

> > > ### Comment · Area_Chair_xkQ5 · 2024-11-26
> > >
> > > Dear reviewer,
> > >
> > > Please make to sure to read, at least acknowledge, and possibly further discuss the authors' responses to your comments. Update or maintain your score as you see fit.
> > >
> > > The AC.

---

> > > > ### Author Response · Authors · 2024-11-29
> > > >
> > > > Dear reviewer,
> > > >
> > > > Just a gentle reminder that the last day to make a comment is December 2nd. If there are any additional questions or concerns we would be happy to address them. Thank you !

---

> > > > > ### Author Response · Authors · 2024-12-01
> > > > >
> > > > > Dear Reviewer 9pTM,
> > > > >
> > > > > Just a friendly reminder if you would like to read the rebuttal or provide any additional response. Thanks!

---

> ### Comment · Reviewer_9pTM · 2024-12-02
> **Thank you for the detailed response**
>
> I would like to thank the authors for the detailed response to my comments. Especially for adding the new dataset as well as the figures for the closest training samples. Please find my responses inline.
>
> > For the new dataset with regular shapes, we have run a new experiment to address this. In particular, we have run experiments and baselines on a 3D turbulence dataset with 45 various shapes (squares, rectangles, steps, bars, elbows, donut, U-shape, T-> shape, H-shape etc.) with highly complex flow patterns, and also manually captioned this dataset which is more amenable to text descriptions. For a full picture of all the shapes, we recommend the last page of Lienen et al. (https://arxiv.org/abs/2306.01776)
>
> Thank you for adding the dataset. I think this would be a great one to showcase the usefulness of text for describing the geometry. Especially, since the new dataset is 3D, it is clearly the most challenging and high signal dataset in this paper. However, I believe (a) the new dataset changes the paper significantly and I would suggest resubmission (b) again, as the main focus is text-based conditioning of the initial and boundary conditions, I still consider that a text2img + neural solver (or even text2img + traditional solver) baselines to be a requirement to ascertain the usefulness of the model.
>
> > Lastly, using text2img coupled with a PDE solver seems quite interesting and could be a direction for future work. However, one reason that we decided on the current method is the training difficulties that could arise when decoupling the solution process. In particular, the generative text2img process seeks to generate an initial condition that matches the description yet this can lead to multiple viable initial flow fields (for example in smoke buoyancy) or even small fluctuations in the sampled initial condition.
>
> Thank you for spelling out the difficulties of the approach! However, it is not clear to me why your current approach doesn't face the same difficulties. Text conditioning is inherently a fuzzy description in the setting of PDEs -- which is why I think describing a smoke plume is not the right dataset for this problem setting. (Again, simple geometric shapes would be much better!). Furthermore, if you do think that decoupling approach would encounter those difficulties, the paper would be strengthened if the baseline is implemented and the shortcoming demonstrated with a expanded discussion of this above paragraph.
>
> I continue to maintain that a decoupled text2img + neural pde solver would be the most important baseline to compare against in this problem setting.
>
> > Thank you for the question, we have anonymously posted an image of the closest training samples to the validation set here, > > since the manuscript has gotten quite long and we are unsure of how to fit this in: https://i.postimg.cc/t4d0jcV7/train-closest.png
> > The training set images are quite close to the generated samples, which in this case seems to be a limitation of the given
> > dataset (from a separate paper) and the captioning process.
> > The validation samples themselves are quite similar to the training set, and using an LLM to caption the two data splits could
> > also result in similarities in the captions as well.
>
> Thank you for adding the closest training sample which matches the generated initial condition. As they are extremely close, this confirms my suspicion that the proposed model is memorizing the training sample rather than generating a new sample. I agree that the dataset choice is limiting for the problem setting.
>
> > However, we can notice that the solutions do diverge over time, and eventually
> > reach noticeably different states. While the initial conditions are quite similar, small deviations eventually cause differences
> > that the model would need to accurately model.
>
> That's a great point. Rollouts of chaotic PDEs do magnify subtle variations in the initial condition. That the small deviations diverge and they are still accurately modeled speaks to the fact that the neural PDE solver part is doing its job. This has been done in prior work and doesn't demonstrate that the text conditioning part is useful.
>
> In conclusion, I would encourage the authors to continue the line of research which combines text conditioning and neural PDE solvers. I urge the authors to carefully choose the datasets which demonstrate the benefits of the text conditioning, as well as add the suggested baseline comparisons. At this time, I have decided to maintain my current scores.

---

> > ### Author Response · Authors · 2024-12-02
> >
> > Thank you for the reply and for the constructive feedback! Unfortunately it seems as if we have run out of time to run the text2img + solver baseline, however it is a great suggestion. Despite not having concrete results, there is some intuition from prior works comparing text2img + temporal rollout/inflation vs. full text2video generation that seem to suggest that the full generation is more temporally consistent [1]. To mitigate this, some auto-regressive text2video generators use an "anchor/reference frame" to ensure the consistency of the auto-regressive generator with respect to the initial condition [2]. Regardless, we are happy to have addressed your other concerns, including the novel evaluation and dataset, and we appreciate the time and effort in providing your feedback.
> >
> > 1. Omer Bar-Tal, et al., Lumiere: A Space-Time Diffusion Model for Video Generation, https://arxiv.org/abs/2401.12945
> > 2. Wenming Weng, et al., ART⋅V: Auto-Regressive Text-to-Video Generation with Diffusion Models, https://arxiv.org/abs/2311.18834

---

### Official Review · Reviewer_tcKA · 2024-11-06

**Soundness:** 2
**Presentation:** 2
**Contribution:** 2
**Rating:** 6
**Confidence:** 3

**Summary:**

this paper introduce framework for text to simulation use latent diffusion models, enables condition on language, initial conditions. this work generate a entire trajectories for improved accuracy without error accumulation.

**Strengths:**

- apply latent diffusion models to physics simulation with mesh encoder/decoder provide a scalable solution.
- a new text conditioned PDE simulation framework.
- supporting both structured and unstructured grids, and shows promising results on different types of PDE problems

**Weaknesses:**

-  it's unclear the benefits and why doing text-conditioned simulation generation
- I find paper's contribution is more on using latent diffusion for pde simulation, not the text to PDE generation.

**Questions:**

- what is re-solved trajectory referred to in line 394?
- why doing text-conditioned simulation generation, is the pde type pre-defined for each model? How important is the training data text caption?
- how does text-conditioning help the simulation results or they are dis-entangled.

---

> ### Author Response · Authors · 2024-11-25
> **Response to Reviewer tcKA**
>
> Thank you for taking the time to write a review. Please see our response and feel free to reach out if there are any further concerns/questions.
>
> >it's unclear the benefits and why doing text-conditioned simulation generation
>
> Thank you for bringing this point up. We believe that neural PDE solvers are becoming increasingly limited by a knowledge/accessibility gap when it comes to their practical application. As technical problems are being solved (more accurate prediction, more generalizable, etc.) the main limitation is increasingly becoming the steep knowledge gap and inaccessibility of novel methods. This can also be seen in the market: new startups leveraging SOTA research generally serve as consultants or service providers to large engineering firms to deliver neural solver solutions due to the unique, interdisciplinary background needed to deploy these models. While this isn’t a purely scientific problem, it is still highly relevant for the practical use of neural solvers.
>
> To frame this in another light, image generation has been possible for many years and at very high quality. However, despite the technology and models being there, image generation was still largely only used within the CV community until text2image models were introduced; it is likely that a main factor for this is image generation models were highly inaccessible to practitioners before the use of a text modality. Artists, advertisers, or even just the general public are now able to leverage deep learning partly due to research targeted at the accessibility of image generation.
>
> >I find paper's contribution is more on using latent diffusion for pde simulation, not the text to PDE generation.
>
> Thank you for raising this concern, and indeed, we had to advance many aspects of latent diffusion for PDEs in order to even begin using them for text-conditioned generation. To the best of our knowledge, this paper is the first to do:
>
> - Full 4D spatiotemporal generation
> - Conditional latent diffusion
> - Diffusion on arbitrary meshes
>
> for PDE problems, allowing diffusion models to become much faster (through a latent space + spatiotemporal generation) and applicable to a broad set of physics scenarios. We believe that these contributions are synergistic with text-conditioned generation and believe we can contribute both to latent diffusion models and text2PDE modeling. Indeed, on the language modeling front we also contribute three new captioned datasets and, for the first time, propose text to physics simulation and demonstrate that it can be done.
>
> >what is re-solved trajectory referred to in line 394?
>
> Thank you for raising this question. We describe this in the paper in Section 4.2 results, but in summary, many initial conditions can satisfy a given text prompt, yet not necessarily match an initial condition in the validation set. Rather than compute errors between samples in the validation set (which would penalize the diversity of the trained generative model), we sample solutions based on a text prompt and use the sampled initial condition to re-solve a trajectory using a numerical solver. This re-solved trajectory is then used to evaluate the loss and measure the physical consistency of the generated solutions.
>
> > why doing text-conditioned simulation generation, is the pde type pre-defined for each model? How important is the training data text caption?
>
> Thank you for the question. There are no limitations on the PDE type that the model can learn from, as long as the PDE can be captured by a dataset. At this stage, we believe the text caption is reasonably important to the model performance; without good text captions the model would be essentially doing unconditional generation. Unfortunately, it is difficult to rigorously measure the sensitivity to input prompts, as it is both subjective and would require large amounts of data captioning, but this is certainly an area for future works to expand on and work with the initial captioned datasets we have released.
>
> > how does text-conditioning help the simulation results or they are dis-entangled.
>
> When used for text-conditioned generation, the model is only given a text prompt along with gaussian noise as an input. Therefore the text-conditioning would be the only information the model has to generate a solution, and can encode the boundary condition, initial condition, and any other relevant information.

---

> > ### Comment · Area_Chair_xkQ5 · 2024-11-26
> >
> > Dear reviewer,
> >
> > Please make to sure to read, at least acknowledge, and possibly further discuss the authors' responses to your comments. Update or maintain your score as you see fit.
> >
> > The AC.

---

> > > ### Author Response · Authors · 2024-11-29
> > >
> > > Dear reviewer,
> > >
> > > Just a gentle reminder that the last day to make a comment is December 2nd. If there are any additional questions or concerns we would be happy to address them. Thank you !

---

### Author Response · Authors · 2024-11-25
**Thank you to all Reviewers**

Thank you to reviewers for writing thoughtful reviews. We have individually replied to each review and updated the manuscript, however for easy reference we have summarized the changes here:

## Major Changes
- Added new 3D Turbulence experiments to investigate scalability + complex systems
    - Section 4.3: Main results + errors
    - Appendix A: New visualizations
    - Appendix E.3: Dataset description
    - Appendix D.3: Captioning description
    - Appendix G: Added inference times
- Added new experiments to allow the LDM framework to also be used auto-regressively to enable scaling to longer time-horizons and multi-step prediction
    - Appendix C.4: Main results + method explanation
- Added new experiments using DDIM to accelerate LDM sampling
    - Appendix C.3: Main results + rationale/explanation

## Minor Changes
- Added more details on the kernel aggregation/interpolation to explain discretization invariance.
    - Appendix B.3: Improved explanation and new visualizations
- Added more details about the Cylinder Flow dataset
    - Appendix E.1: Improved explanation and new visualizations
- Changed L1 Loss to Relative L2 Loss in main text tables
    - Tables 1 and 2: Changed to L2 loss
- Added ablation study on ball radius
    - Appendix B.1: New table and discussion
- Clarified Cylinder Flow captioning
    - Appendix D.1: minor clarifications

We encourage reviewers to browse the updated manuscript due to the large number of changes. Additionally, we apologize for the late reply, however, we are still interested if there are any additional comments or thoughts. Thank you!

---

### Comment · Area_Chair_xkQ5 · 2024-11-26

Dear all,

The deadline for the authors-reviewers phase is approaching (December 2).

@For reviewers, please read, acknowledge and possibly further discuss the authors' responses to your comments. While decisions do not need to be made at this stage, please make sure to reevaluate your score in light of the authors' responses and of the discussion.

- You can increase your score if you feel that the authors have addressed your concerns and the paper is now stronger.
- You can decrease your score if you have new concerns that have not been addressed by the authors.
- You can keep your score if you feel that the authors have not addressed your concerns or that remaining concerns are critical.

Importantly, you are not expected to update your score. Nevertheless, to reach fair and informed decisions, you should make sure that your score reflects the quality of the paper as you see it now. Your review (either positive or negative) should be based on factual arguments rather than opinions. In particular, if the authors have successfully answered most of your initial concerns, your score should reflect this, as it otherwise means that your initial score was not entirely grounded by the arguments you provided in your review. Ponder whether the paper makes valuable scientific contributions from which the ICLR community could benefit, over subjective preferences or unreasonable expectations.

@For authors, please respond to remaining concerns and questions raised by the reviewers. Make sure to provide short and clear answers. If needed, you can also update the PDF of the paper to reflect changes in the text. Please note however that reviewers are not expected to re-review the paper, so your response should ideally be self-contained.

The AC.

---

### Meta-Review · Area_Chair_xkQ5 · 2024-12-21

**Metareview:**

The reviewers are divided (6-5-6-6-6-3) about the paper, but they overall lean towards acceptance. The paper introduces a new approach to physics simulations using latent diffusion models that can be conditioned on text prompts. Along the way, the paper introduces full 4d spatiotemporal generation and diffusion on arbitrary (non-regular) meshes. A number of concerns have been raised by the reviewers, and most of them have been addressed during the author-reviewer discussion period. The proposed changes include clarifications and new experiments that better investigate Text2PDE's capabilities.

The main concern that remains is with regards to the text conditioning capability of Text2PDE. Conditioning on text prompts is an interesting direction but makes the evaluation more challenging, as text descriptions are inherently fuzzy. To address this issue, a new dataset with 3d shapes has been proposed by the authors.

Overall, while the usefulness and validation of text conditioning may still be debated, the paper presents valuable contributions in the context of physics simulations using latent diffusion models. For these reasons, and given the overall positive reviews, I recommend acceptance. I encourage the authors to address the remaining concerns and to implement the modifications discussed with the reviewers in the final version of the paper.

**Additional Comments On Reviewer Discussion:**

A number of concerns have been raised by the reviewers, and most of them have been addressed during the author-reviewer discussion period. The proposed changes include clarifications and new experiments that better investigate Text2PDE's capabilities.

---

### Decision · Program_Chairs · 2025-01-22

Accept (Poster)